# Endotaxis: A neuromorphic algorithm for mapping, goal-learning, navigation, and patrolling

**Tony Zhang[1], Matthew Rosenberg[1,2], Zeyu Jing[1], Pietro Perona[3], Markus Meister[1]***

[1]Division of Biology and Biological Engineering, California Institute of Technology, Pasadena, United States; [2]Center for the Physics of Biological Function, Princeton University, Princeton, United States; [3]Division of Engineering and Applied Science, California Institute of Technology, Pasadena, United States

**Abstract** An animal entering a new environment typically faces three challenges: explore the space for resources, memorize their locations, and navigate towards those targets as needed. Here we propose a neural algorithm that can solve all these problems and operates reliably in diverse and complex environments. At its core, the mechanism makes use of a behavioral module common to all motile animals, namely the ability to follow an odor to its source. We show how the brain can learn to generate internal "virtual odors" that guide the animal to any location of interest. This *endotaxis* algorithm can be implemented with a simple 3-layer neural circuit using only biologically realistic structures and learning rules. Several neural components of this scheme are found in brains from insects to humans. Nature may have evolved a general mechanism for search and navigation on the ancient backbone of chemotaxis.

**\*For correspondence:**
meister4@mac.com

## eLife assessment

This **valuable** work proposes a framework inspired by chemotaxis for understanding how the brain might implement behaviors related to navigating toward a goal. The evidence supporting the conceptual claim is **convincing**. The article proposes a hypothesis that would be of interest to the broad systems neuroscience community, although it was noted the relationship to existing similar hypotheses could be clarified.

## Introduction

Animals navigate their environment to look for resources – such as shelter, food, or a mate – and exploit such resources once they are found. Efficient navigation requires knowing the structure of the environment: which locations are connected to which others (*Tolman, 1948*). One would like to understand how the brain acquires that knowledge, what neural representation it adopts for the resulting map, how it tags significant locations in that map, and how that knowledge gets read out for decision-making during navigation.

Experimental work on these topics has mostly focused on simple environments – such as an open arena (*Wilson and McNaughton, 1993*), a pond (*Morris et al., 1982*), or a desert (*Müller and Wehner, 1988*) – and much has been learned about neural signals in diverse brain areas under these conditions (*Sosa and Giocomo, 2021*; *Collett and Collett, 2002*). However, many natural environments are highly structured, such as a system of burrows, or of intersecting paths through the underbrush. Similarly, for many cognitive tasks, a sequence of simple actions can give rise to complex solutions.

One algorithm for finding a valuable resource is common to all animals: chemotaxis. Every motile species has a way to track odors through the environment, either to find the source of the odor or to avoid it (*Baker et al., 2018*). This ability is central to finding food, connecting with a mate, and avoiding predators. It is believed that brains originally evolved to organize the motor response in pursuit of chemical stimuli. Indeed, some of the oldest regions of the mammalian brain, including the hippocampus, seem organized around an axis that processes smells (*Jacobs, 2012*; *Aboitiz and Montiel, 2015*).

The specifics of chemotaxis, namely the methods for finding an odor and tracking it, vary by species, but the toolkit always includes a search strategy based on trial-and-error: try various actions that you have available, then settle on the one that makes the odor stronger (*Baker et al., 2018*). For example, a rodent will weave its head side-to-side, sampling the local odor gradient, then move in the direction where the smell is stronger. Worms and maggots follow the same strategy. Dogs track a ground-borne odor trail by casting across it side-to-side. Flying insects perform similar casting flights. Bacteria randomly change direction every now and then, and continue straight as long as the odor improves (*Berg, 1988*). We propose that this universal behavioral module for chemotaxis can be harnessed to solve general problems of search and navigation in a complex environment, even when tell-tale odors are not available.

For concreteness, consider a mouse exploring a labyrinth of tunnels (*Figure 1A*). The maze may contain a source of food that emits an odor (*Figure 1A1*). That odor will be strongest at the source and decline with distance along the tunnels of the maze. The mouse can navigate to the food location by simply following the odor gradient uphill. Suppose that the mouse discovers some other interesting locations that *do not* emit a smell, like a source of water, or the exit from the labyrinth (*Figures 1A2–3*). It would be convenient if the mouse could tag such a location with an odorous material, so it may be found easily on future occasions. Ideally, the mouse would carry with it multiple such odor tags, so it can mark different targets each with its specific recognizable odor.

Here we show that such tagging does not need to be physical. Instead, we propose a mechanism by which the mouse's brain may compute a 'virtual odor' signal that declines with distance from a chosen target. That neural signal can be made available to the chemotaxis module as though it were a real odor, enabling navigation up the gradient toward the target. Because this goal signal is computed in the brain rather than sensed externally, we call this hypothetical process *endotaxis*.

The developments reported here were inspired by a recent experimental study with mice navigating a complex labyrinth (*Rosenberg et al., 2021*) that includes 63 three-way junctions. Among other things, we observed that mice could learn the location of a resource in the labyrinth after encountering it just once, and perfect a direct route to that target location after $\sim 10$ encounters. Furthermore, they could navigate back out of the labyrinth using a direct route they had not traveled before, even on the first attempt. Finally, the animals spent most of their waking time patrolling the labyrinth, even long after they had perfected the routes to rewarding locations. These patrols covered the environment efficiently, avoiding repeat visits to the same location. All this happened within a few hours of the animal's first encounter with the labyrinth. Our modeling efforts here are aimed at explaining these remarkable phenomena of rapid spatial learning in a new environment: one-shot learning of a goal location, zero-shot learning of a return route, and efficient patrolling of a complex maze. In particular we want to do so with a biologically plausible mechanism that could be built out of neurons.

## Results

### A neural circuit to implement endotaxis

*Figure 1B* presents a neural circuit model that implements three goals: mapping the connectivity of the environment; tagging of goal locations with a virtual odor; and navigation toward those goals. The model includes four types of neurons: resource cells, point cells, map cells, and goal cells.

### Resource cells

These are sensory neurons that fire when the animal encounters an interesting resource, for example, water or food, that may form a target for future navigation. Each resource cell is selective for a specific kind of stimulus. The circuitry that produces these responses is not part of the model.

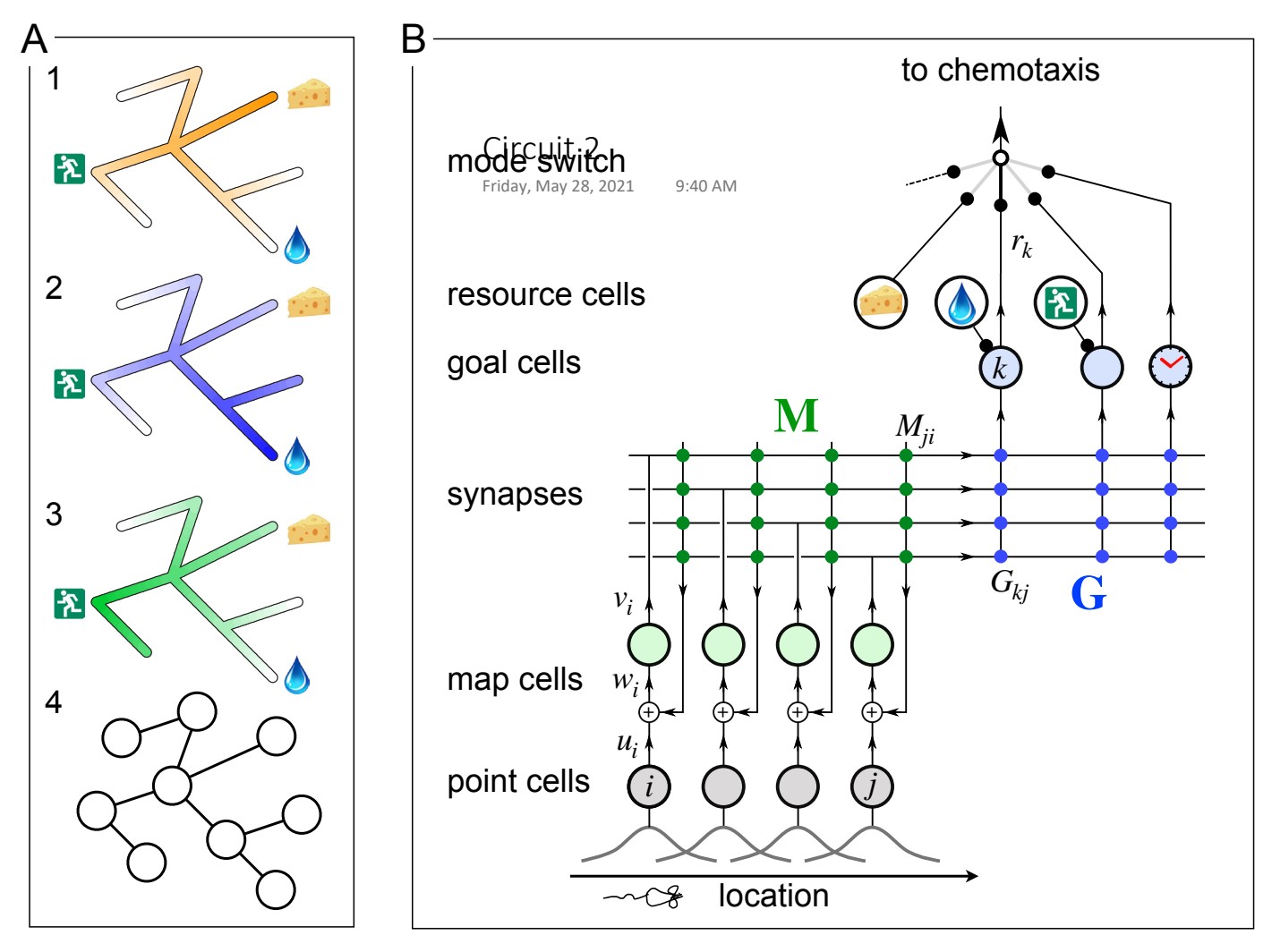

**Figure 1.** A mechanism for endotaxis. (**A**) A constrained environment of tunnels linked by intersections, with special locations offering food, water, and the exit. (1) A real odor emitted by the food source decreases with distance (shading). (2) A virtual odor tagged to the water source. (3) A virtual odor tagged to the exit. (4) Abstract representation of this environment by a graph of nodes (intersections) and edges (tunnels). (**B**) A neural circuit to implement endotaxis. Open circles: four populations of neurons that represent 'resource,' 'point,' 'map,' and 'goal.' Arrows: signal flow. Solid circles: synapses. Point cells have small receptive fields localized in the environment and excite map cells. Map cells excite each other (green synapses) and also excite goal cells (blue synapses). Resource cells signal the presence of a resource, for example, cheese, water, or the exit. Map synapses and goal synapses are modified by activity-dependent plasticity. A 'mode' switch selects among various goal signals depending on the animal's need. They may be virtual odors (water, exit) or real odors (cheese). Another goal cell (clock) may report how recently the agent has visited a location. The output of the mode switch gets fed to the chemotaxis module for gradient ascent. Mathematical symbols used in the text: $u_i$ is the output of a point cell at location $i$, $w_i$ is the input to the corresponding map cell, $v_i$ is the output of that map cell, $\mathbf{M}$ is the matrix of synaptic weights among map cells, $\mathbf{G}$ are the synaptic weights from the map cells onto goal cells, and $r_k$ is the output of goal cell $k$.

### Point cells

This layer of cells represents the animal's location. (We avoid the term 'place cell' here because [1] that term has a technical meaning in the rodent hippocampus, whereas the arguments here extend to species that do not have a hippocampus; and [2] all the cells in this network have a place field, but it is smallest for the point cells.) Each neuron in this population has a small response field within the environment. The neuron fires when the animal enters that response field. We assume that these point cells exist from the outset as soon as the animal enters the environment. Each cell's response field is defined by some conjunction of external and internal sensory signals at that location.

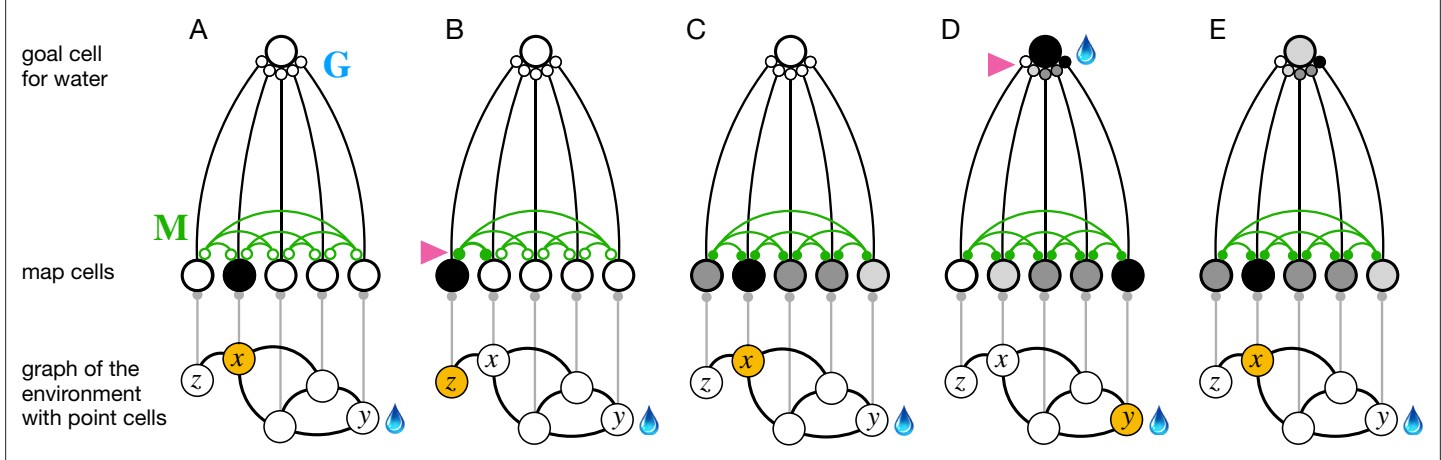

**Figure 2.** The phases of endotaxis during exploration, goal-tagging, and navigation. A portion of the circuit in *Figure 1* is shown, including a single goal cell that responds to the water resource. Bottom shows a graph of the environment, with nodes linked by edges, and the agent's current location shaded in orange. Each node has a point cell that reports the presence of the agent to a corresponding map cell. Map cells are recurrently connected (green) and feed convergent signals onto the goal cell. (**A**) Initially the recurrent synapses are weak (empty circles). (**B**) During exploration, the agent moves between two adjacent nodes on the graph, and that strengthens (arrowhead) the connection between their corresponding map cells (filled circles). (**C**) After exploration, the map synapses reflect the connectivity of the graph. Now the map cells have an extended profile of activity (darker = more active), centered on the agent's current location $x$ and decreasing from there with distance on the graph. (**D**) When the agent reaches the water source $y$, the goal cell gets activated by the sensation of water, and this triggers plasticity (arrowhead) at its input synapses. Thus, the state of the map at the water location gets stored in the goal synapses. This event represents tagging of the water location. (**E**) During navigation, as the agent visits different nodes, the map state gets filtered through the goal synapses to excite the goal cell. This produces a signal in the goal cell that declines with the agent's distance from the water location.

## Map cells

This layer of neurons learns the structure of the environment, namely how the various locations are connected in space. The map cells get excitatory input from point cells in a one-to-one fashion. These input synapses are static. The map cells also excite each other with all-to-all connections. These recurrent synapses are modifiable according to a local plasticity rule. After learning, they represent the topology of the environment.

## Goal cells

Each goal cell serves to mark the locations of a special resource in the map of the environment. The goal cell receives excitatory input from a resource cell, which gets activated whenever that resource is present. It also receives excitatory synapses from map cells. Such a synapse is strengthened when the presynaptic map cell is active at the same time as the resource cell.

After the map and goal synapses have been learned, each goal cell carries a virtual odor signal for its assigned resource. The signal increases systematically as the animal moves closer to a location with that resource. A mode switch selects one among many possible virtual odors (or real odors) to be routed to the chemotaxis module for odor tracking. (The mode switch effectively determines the animal's behavioral policy. In this report, we do not consider how or why the animal chooses one mode or another.) The animal then pursues its chemotaxis search strategy to maximize that odor, which leads it to the selected tagged location.

## Why does the circuit work?

The key insight is that the output of the goal cell declines systematically with the distance of the animal from the target location. This relationship holds even if the environment is constrained with a complex connectivity graph (*Figure 1A4*). Here we explain how this comes about, with mathematical details to follow.

In a first phase, the animal explores the environment while the circuit builds a map. When the animal moves from one location to an adjacent one, those two point cells fire in rapid succession. That leads to a Hebbian strengthening of the excitatory synapses between the two corresponding map

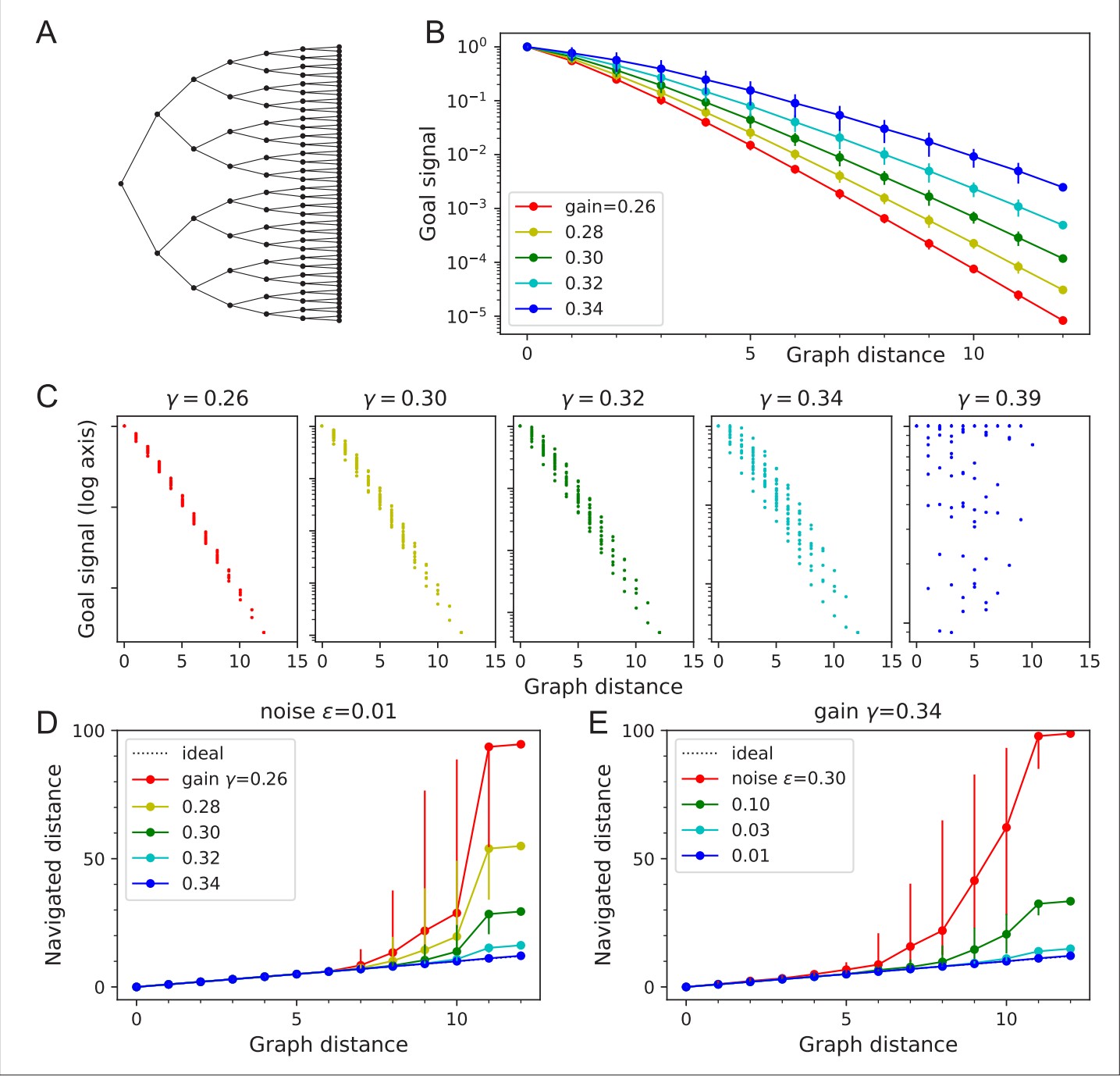

**Figure 3.** Theory of the goal signal. Dependence of the goal signal on graph distance, and the consequences for endotaxis navigation. (**A**) The graph representing a binary tree labyrinth (**Rosenberg et al., 2021**) serves for illustration. Suppose the endotaxis model has acquired the adjacency matrix perfectly: $\mathbf{M} = \mathbf{A}$. We compute the goal signal $E_{xy}$ between any two nodes on the graph and compare the results at different values of the map gain $\gamma$. (**B**) Dependence of the goal signal $E_{xy}$ on the graph distance $D_{xy}$ between the two nodes. Mean ± SD, error bars often smaller than markers. The maximal distance on this graph is 12. Note logarithmic vertical axis. The signal decays exponentially over many log units. At high $\gamma$, the decay distance is greater. (**C**) A detailed look at the goal signal, each point is for a pair of nodes $(x, y)$. For low $\gamma$, the decay with distance is strictly monotonic. At high $\gamma$, there is overlap between the values at different distances. As $\gamma$ exceeds the critical value $\gamma_c = 0.38$, the distance dependence breaks down. (**D**) Using the goal signal for navigation. For every pair of start and end nodes, we navigate the route by following the goal signal and compare the distance traveled to the shortest graph distance. For all routes with the same graph distance, we plot the median navigated distance with 10 and 90% quantiles. Variable gain at a constant noise value of $\epsilon = 0.01$. (**E**) As in panel (**D**) but varying the noise at a constant gain of $\gamma = 0.34$.

cells (*Figure 2A and B*). In this way, the recurrent network of map cells learns the connectivity of the graph that describes the environment. To a first approximation, the matrix of synaptic connections among the map cells will converge to the correlation matrix of their inputs (*Dayan and Abbott, 2001*; *Galtier et al., 2012*), which in turn reflects the adjacency matrix of the graph (*Equation 1*). Now the brain can use this adjacency information to find the shortest path to a target.

After this map learning, the output of the map network is a hump of activity, centered on the current location $x$ of the animal and declining with distance along the various paths in the graph of the environment (*Figure 2C*). If the animal moves to a different location $y$, the map output will change to another hump of activity, now centered on $y$ (*Figure 2D*). The overlap of the two hump-shaped profiles will be large if nodes $x$ and $y$ are close on the graph, and small if they are distant. Fundamentally the endotaxis network computes that overlap.

Suppose the animal visits $y$ and finds water there. Then the water resource cell fires, triggering synaptic learning in the goal synapses. That stores the current profile of map activity $v_i(y)$ in the synapses $G_{ki}$ onto the goal cell $k$ that responds to water (*Figure 2D*, *Equation 9*). When the animal subsequently moves to a different location $x$, the goal cell $k$ receives the current map output $\mathbf{v}(x)$ filtered through the previously stored synaptic template $\mathbf{v}(y)$ (*Figure 2E*). This is the desired measure of overlap (*Equation 10*). Under suitable conditions, this goal signal declines monotonically with the shortest graph distance between $x$ and $y$, as we will demonstrate both analytically and in simulations (sections 'Theory of endotaxis' and 'Acquisition of map and targets during exploration').

## Theory of endotaxis

Here we formalize the processes of *Figure 2* in a concrete mathematical model. The model is simple enough to allow some exact predictions for its behavior. The present section develops an analytical understanding of endotaxis that will help guide the numerical simulations in subsequent parts.

The environment is modeled as a graph consisting of $n$ nodes, with adjacency matrix

$$A_{ij} = \begin{cases} 1, & \text{if node } i \text{ can be reached from node } j \text{ in one step} \\ 0, & \text{otherwise, including the } i = j \text{ case} \end{cases} \tag{1}$$

We suppose the graph is undirected, meaning that every link can be traversed in both directions,

$$A_{ij} = A_{ji}$$

Movements of the agent are modeled as a sequence of steps along that graph. During exploration, the agent performs a walk that tries to cover the entire environment. In the process, it learns the adjacency matrix $\mathbf{A}$. During navigation, the agent uses that knowledge to travel to a known target.

For an agent navigating a graph, it is very useful to know the shortest graph distance between any two nodes

$$D_{ij} = \text{minimum number of steps needed to reach node } i \text{ from node } j \tag{2}$$

Given this information, one can navigate the shortest route from $x$ to $y$: for each of the neighbors of $x$, look up its distance to $y$ and step to the neighbor with the shortest distance. Then repeat that process until $y$ is reached. Thus, the shortest route can be navigated one step at a time without any high-level advanced planning. This is the core idea behind endotaxis.

The network of *Figure 1B* effectively computes the shortest graph distances. We implement the circuit as a textbook linear rate model (*Dayan and Abbott, 2001*). Each map unit $i$ has a synaptic input $w_i$ that it converts to an output $v_i$,

$$v_i = \gamma w_i \tag{3}$$

where $\gamma$ is the gain of the units. The input consists of an external signal $u_i$ summed with a recurrent feedback through a connection matrix $\mathbf{M}$

$$w_i = u_i + \sum_{ij} M_{ij} v_j \tag{4}$$

where $M_{ij}$ is the synaptic strength from unit $j$ to $i$.

The point neurons are one-hot encoders of location. A point neuron fires if the agent is at that location; all the others are silent:

$$u_i(x) = \text{firing rate of point cell } i \text{ with the agent at node } x$$
$$= \delta_{ix} \tag{5}$$

where $\delta_{ix}$ is the Kronecker delta.

So the vector of all map outputs is

$$\mathbf{v} = \gamma \left( \mathbf{u} + \mathbf{M}\mathbf{v} \right) = \left( \frac{1}{\gamma}\mathbf{1} - \mathbf{M} \right)^{-1} \mathbf{u} \tag{6}$$

where $\mathbf{u}$ is the one-hot input from point cells.

Now consider goal cell number $k$ that is associated to a particular location $y$ because its resource is present at that node. The goal cell sums input from all the map units $v_i$, weighted by its goal synapses $G_{ki}$. So with the agent at node $x$, the goal signal $r_k$ is

$$r_k(x) = \sum_i G_{ki} \cdot v_i(x) = \mathbf{g}_k \cdot \mathbf{v}(x) = \mathbf{g}_k \cdot \left( \frac{1}{\gamma}\mathbf{1} - \mathbf{M} \right)^{-1} \mathbf{u}(x) \tag{7}$$

where we write $\mathbf{g}_k$ for the $k$th row vector of the goal synapse matrix $\mathbf{G}$. This is the set of synapses from all map cells onto the specific goal cell in question.

Suppose now that the agent has learned the structure of the environment perfectly, such that the map synapses are a copy of the graph's adjacency matrix (1),

$$\mathbf{M} = \mathbf{A} \tag{8}$$

Similarly, suppose that the agent has acquired the goal synapses perfectly, namely proportional to the map output at the goal location $y$:

$$\mathbf{g}_k = \mathbf{v}(y) \tag{9}$$

Then as the agent moves to another location $x$, the goal cell reports a signal

$$r_k(x) = \mathbf{g}_k \cdot \mathbf{v}(x) = \mathbf{v}(y) \cdot \mathbf{v}(x) \equiv E_{xy} \tag{10}$$

where the matrix

$$\mathbf{E} = \left( \frac{1}{\gamma}\mathbf{1} - \mathbf{A} \right)^{-1^\top} \left( \frac{1}{\gamma}\mathbf{1} - \mathbf{A} \right)^{-1} \tag{11}$$

It has been shown (**Meister, 2023**) that for small values of $\gamma$ the elements of the resolvent matrix

$$\mathbf{Y} = \left( \frac{1}{\gamma}\mathbf{1} - \mathbf{A} \right)^{-1} \tag{12}$$

are monotonically related to the shortest graph distances $\mathbf{D}$. Specifically,

$$Y_{xy} \xrightarrow[\gamma \to 0]{} \gamma^{1+D_{xy}} \tag{13}$$

Building on that, the matrix $\mathbf{E}$ becomes

$$E_{xy} \xrightarrow[\gamma \to 0]{} \sum_z \gamma^{1+D_{zx}} \gamma^{1+D_{zy}} = \sum_z \gamma^{2+D_{zx}+D_{zy}} \tag{14}$$

The limit is dominated by the term with the smallest exponent, which occurs when $z$ lies on a shortest path from $x$ to $y$

$$\min_z(D_{zx} + D_{zy}) = D_{xy}$$

where we have used the undirected nature of the graph, namely $D_{zx} = D_{xz}$.

Therefore,

$$E_{xy} \xrightarrow[\gamma \to 0]{} \gamma^{2+D_{xy}} \tag{15}$$

where $D_{xy}$ is the smallest number of steps needed to get from node $y$ to node $x$.

*Figure 3* illustrates this relationship with numerical results on a binary tree graph. As expected, for small $\gamma$ the goal signal decays exponentially with graph distance (*Figure 3B*). Therefore, an agent that makes local turning decisions to maximize that goal signal will reach the goal by the shortest possible path.

The exponential decay of the goal signal represents a challenge for practical implementation with biological circuits. Neurons have a finite signal-to-noise ratio, so detecting minute differences in the firing rate of a goal neuron will be unreliable. Because the goal signal changes by a factor of $\gamma$ across every link in the graph, one wants to set the map neuron gain $\gamma$ as large as possible. However, there is a *critical gain* value $\gamma_c$ that sets a strict upper limit:

$$\gamma < \gamma_c \equiv \frac{1}{\text{largest absolute eigenvalue of } \mathbf{A}} \tag{16}$$

For larger $\gamma > \gamma_c$, the goal signal $E_{xy}$ no longer represents graph distances (*Meister, 2023*). The largest eigenvalue of the adjacency matrix in turn is related to the number of edges per node. For graphs with 2–4 edges per node, $\gamma_c$ is typically about 0.3. The graph in *Figure 3A* has $\gamma_c \approx 0.383$, and indeed $E_{xy}$ becomes erratic as $\gamma$ approaches that value (*Figure 3C*).

To implement the finite dynamic range explicitly, we add some noise to the goal signal of *Equation 11*:

$$r_k(x) = \mathbf{g}_k \cdot \mathbf{v}(x) + \eta \tag{17}$$

where the noise $\eta$ has a Gaussian distribution with full width $\epsilon$:

$$\eta \sim \mathcal{N}(0, (\epsilon/2)^2) \tag{18}$$

The scale $\epsilon$ of this noise is expressed relative to the maximum value of the goal signal. If the agent must decide between two goal signals separated by less than $\epsilon$, the noise will take a toll on the resulting navigation performance.

Of course, neurons everywhere within the network will carry some noise. We lump the cumulative effects of that into the final readout step because that allows for efficient calculations (see section 'Average navigated distance'). (In the circuit of *Figure 1B*, one can envision that the readout noise gets added after the mode switch.) What is a reasonable value for this effective readout noise? For reference, humans and animals can routinely discriminate sensory stimuli that differ by only 1%, for example, the pitch of tones or the intensity of a light, especially if they occur in close succession. Clearly the neurons all the way from receptors to perception must represent those small differences. Thus, we will use $\epsilon = 0.01$ as a reference noise value in many of the results presented here.

The process of navigation toward a chosen goal signal is formalized in Algorithm 1. At each node, the agent inspects the goal signal that would be obtained at all the neighboring nodes, corrupted by the readout noise $\eta$. Then it steps to the neighbor with the highest value. Suppose the agent starts at node $x$ and navigates following the goal signal for node $y$. The resulting navigation route $x = s_0, s_1, \ldots, s_n = y$ has $L_{xy} = n$ steps. Navigation is perfect if this equals the shortest graph distance, $L_{xy} = D_{xy}$. We will assess deviations from perfect performance by the excess length of the routes.

---

Algorithm 1 Navigation.

---

Parameters: gain $\gamma$, noise $\epsilon$
Input: map synapse matrix $\mathbf{M}$, goal synapse vector $\mathbf{g}$
 $s \leftarrow x$ ▷ start navigation at node $x$
 **while** not at goal **do** ▷ stop when goal node is found
 **for** all nodes $j$ that neighbor $s$ **do**
 $\mathbf{u}(j)_i \leftarrow \delta_{i,j}$ for every point cell $i$ ▷ point cell output with agent at node $j$

$$\mathbf{v}(j) \leftarrow \left(\frac{1}{\gamma}\mathbf{1} - \mathbf{M}\right)^{-1}\mathbf{u}(j) \qquad\qquad\qquad \triangleright \text{ map output}$$

 $r(j) \leftarrow \mathbf{g}\,\mathbf{v}(j) + \eta(j)$ ▷ noisy goal signal, $\eta \sim \mathcal{N}(0, (\epsilon/2)^2)$
 **end for**
 $s \leftarrow \arg\max\limits_{j} r(j)$ ▷ choose the neighbor node with the highest goal signal
 **end while**

---

*Figure 3D and E* illustrate how the navigated path distance $L_{xy}$ depends on the noise level $\epsilon$ and the gain $\gamma$. For small gain or high noise, the goal signal extends only over a graph distance of 5–6 links. Beyond that, the navigated distance $L_{xy}$ begins to exceed the graph distance $D_{xy}$. As the gain increases, the goal signal extends further through the graph and navigation becomes reliable over longer distances (*Figure 3D*). Eventually, however, the goal signal loses its monotonic distance dependence (*Figure 3C*). At that stage, navigation across the graph may fail because the agent gets trapped in a local maximum of the goal signal. This can happen even before the critical gain value is reached (*Figure 3C*). For the example in *Figure 3*, the highest useful gain is $\gamma = 0.34$ whereas $\gamma_c = 0.383$.

For any given value of the gain, navigation improves with lower noise levels, as expected (*Figure 3E*). At the reference value of $\epsilon = 0.01$, navigation is perfect even across the 12 links that separate the most distant points on this graph.

In summary, this analysis spells out the challenges that need to be met for endotaxis to work properly. First, during the learning phase, the agent must reliably extract the adjacency matrix of the graph and copy it into its map synapses. Second, during the navigation phase, the agent must evaluate the goal signal with enough resolution to distinguish the values at alternative nodes. The neuronal gain $\gamma$ plays a central role: with $\gamma$ too small, the goal signal decays rapidly with distance and vanishes into the noise just a few steps away from the goal. But at large $\gamma$ the network computation becomes unstable.

## Acquisition of map and targets during exploration

As discussed above, the goal of learning during exploration is that the agent acquires a copy of the graph's adjacency matrix in its map synapses, $\mathbf{M} \approx \mathbf{A}$, and stores the map output at a goal location $y$ in the goal synapses $\mathbf{g} \approx \mathbf{v}(y)$. Here we explore how the rules for synaptic plasticity in the map and goal networks allow that to happen. Algorithm 2 spells out the procedure we implemented for learning from a random walk through the environment.

---

Algorithm 2 Map and goal-learning.

---

Parameters: $\gamma, \theta, \alpha$
Input: adjacency matrix $\mathbf{A}$, resource signals $\mathbf{F}$

 $\mathbf{M} \leftarrow 0$                    ▷ initiate map synapses at 0
 $\mathbf{G} \leftarrow 0$                    ▷ initiate goal synapses at 0
 $t \leftarrow 0$                       ▷ $t$ counts the steps
 $s(t) \leftarrow x$                    ▷ start random walk at $x$
 **while** learning **do**
  $t \leftarrow t + 1$
  $s(t) \leftarrow$ a random neighbor of $s(t-1)$          ▷ continue the random walk
  $u_i(t) \leftarrow \delta_{i,s(t)}$ for every point cell $i$            ▷ point cell output

$$\mathbf{v}(t) \leftarrow \left(\frac{1}{\gamma}\mathbf{1} - \mathbf{M}\right)^{-1}\mathbf{u}(t) \qquad\qquad\qquad ▷ \text{map cell output}$$

  **for** all map cell pairs $(i,j)$ **do**
   **if** $v_j(t-1) > \theta$ and $v_i(t) > \theta$ **then**     ▷ threshold on pre- and post-synaptic activity
    $M_{ji}, M_{ij} \leftarrow 1$         ▷ on a directed graph only increment $M_{ij}$
   **end if**
  **end for**
  $\mathbf{r} \leftarrow \mathbf{G}\mathbf{v}(t)$                  ▷ goal signals
  **for** every goal neuron $k$ **do**
   **if** $F_{k,s(t)} > 0$ **then**      ▷ the agent is at a location that contains resource $k$
    **for** every map neuron $j$ **do**
     $G_{kj} \leftarrow G_{kj} + \alpha(F_{k,s(t)} - r_k)v_j(t)$       ▷ update goal synapses
    **end for**
   **end if**
  **end for**
 **end while**

---

The map synapses $M_{ij}$ start out at zero strength. When the agent moves from node $j = s(t)$ at time $t$ to node $i = s(t+1)$, the map cells $j$ and $i$ are excited in close succession. When that happens, the agent potentiates the synapses between those two neurons to $M_{ji} = M_{ij} = 1$. Of course, a map cell can also get activated through the recurrent network, and we must distinguish that from direct input from its point cell. We found that a simple threshold criterion is sufficient. Here $\theta$ is a threshold applied to both the pre- and postsynaptic activity, and the map synapse gets established only if both neurons respond above threshold. The tuning requirements for this threshold are discussed below.

The goal synapses $G_{kj}$ similarly start out at zero strength. Consider a particular goal cell $k$, and suppose its corresponding resource cell has activity $F_{ky}$ when the agent is at location $y$. When a positive resource signal arrives, that means the agent is at a goal location. If the goal signal $r_k$ received from the map output is smaller than the resource signal $F_{ky}$, then the goal synapses get incremented by something proportional to the current map output. Learning at the goal synapses saturates when the goal signal correctly predicts the resource signal. The learning rate $\alpha$ sets how fast that will happen. Note that both the learning rules for map and goal synapses are Hebbian and strictly local: each synapse is modified based only on signals available in the pre- and postsynaptic neurons.

To illustrate the process of map and goal-learning, we simulate an agent exploring a simple ring graph by a random walk (*Figure 4*). At first, there are no targets in the environment that can deliver a resource (*Figure 4A*). Then we add one target location, and later a second one. Finally, we add a new link to the graph that makes a connection clear across the environment. As the agent explores the graph, we will track how its representations evolve by monitoring the map synapses and the profile of the goal signal.

At the outset, every time the agent steps to a new node, the map synapse corresponding to that link gets potentiated (*Figure 4B*). After enough steps, the agent has executed every link on the graph, and the matrix of map synapses resembles the full adjacency matrix of the graph (*Figure 4B*). At this stage, the agent has learned the connectivity of the environment.

Once a target appears in the environment, it takes the agent a few random steps to encounter it. At that moment, the goal synapses get potentiated for the first time, and suddenly a goal signal appears in the goal cell (*Figure 4C*). The profile of that goal signal is fully formed and spreads through the entire graph thanks to the pre-established map network. By following this goal signal uphill, the agent can navigate along the shortest path to the target from any node on the graph. Note that the absolute scale of the goal signal grows a little every time the agent visits the goal (*Figure 4A*) and eventually saturates.

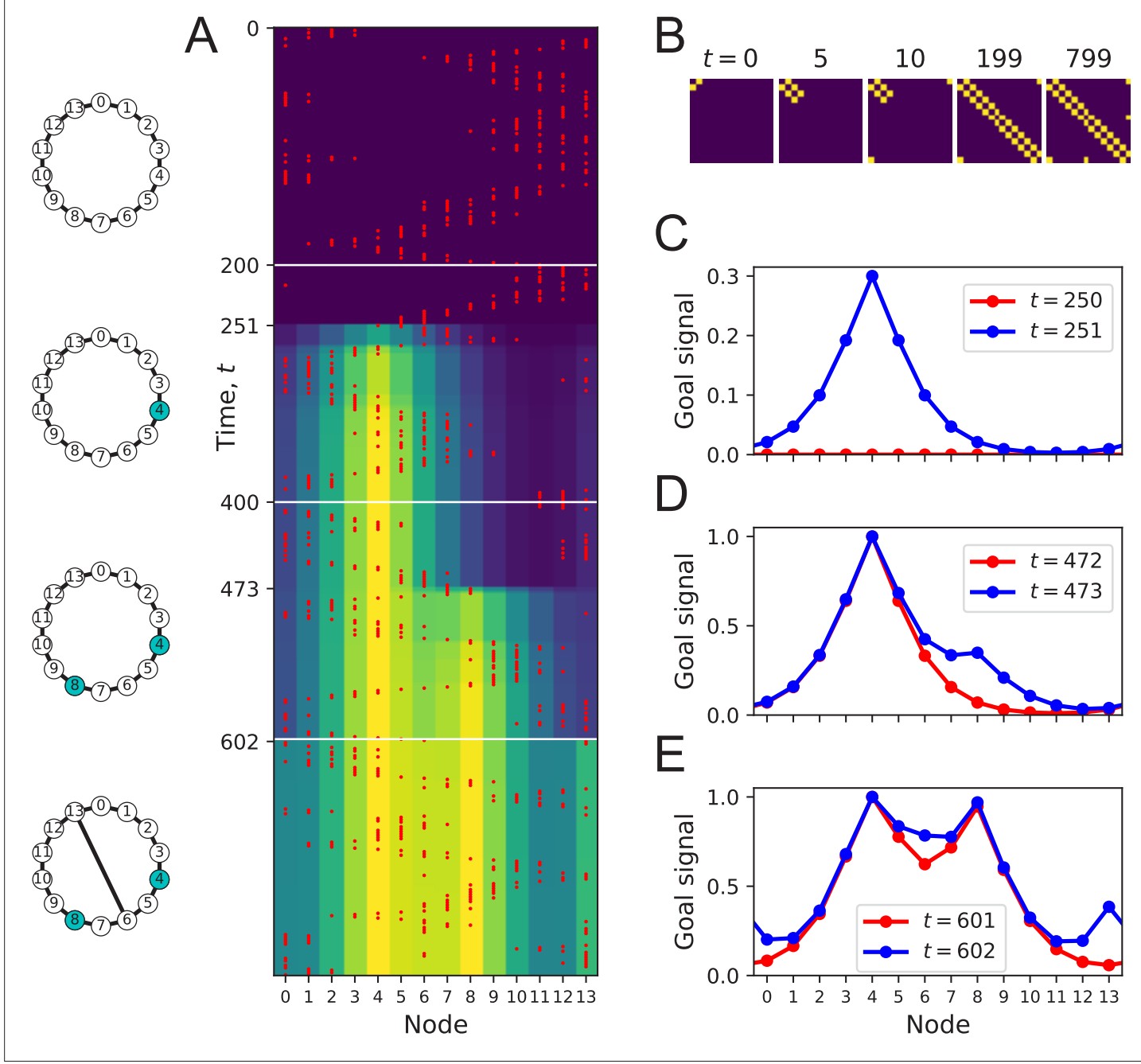

**Figure 4.** Learning the map and the targets during exploration. (**A**) Simulation of a random walk on a ring with 14 nodes. Left: layout of the ring, with resource locations marked in blue. The walk progresses in 800 time steps (top to bottom); with the agent's position marked in red (nodes 0–13, horizontal axis). At each time, the color map shows the goal signal that would be produced if the agent were at position 'Node.' White horizontal lines mark the appearance of a target at $t = 200$, a second target with the same resource at $t = 400$, and a new link across the ring at step $t = 600$. (**B**) The matrix **M** of map synapses at various times. The pixel in row $i$ and column $j$ represents the matrix element $M_{ij}$. Color purple =0. Note the first few steps (number above graph) each add a new synapse. Eventually, **M** reflects the adjacency matrix of nodes on the graph. (**C**) Goal signals just before and just after the agent encounters the first target. (**D**) Goal signals just before and just after the agent encounters the second target. (**E**) Goal signals just before and just after the agent travels the new link for the first time. Parameters: $\gamma = 0.32, \theta = 0.27, \alpha = 0.3$.

Sometime later, we introduce a second target elsewhere in the environment (*Figure 4D*). When the agent encounters it along its random walk, the goal synapses get updated, and the new goal signal has two peaks in its profile. Again, this goal signal grows during subsequent visits. By following that signal uphill from any starting point, the agent will be led to a nearby target by the shortest possible path.

When a new link appears, the agent eventually discovers it on its random walk. At that point, the goal signal changes instantaneously to incorporate the new route (*Figure 4E*). An agent following the new goal signal from node 13 on the ring will now be led to a target location in just three steps, using the shortcut, whereas previously it took five steps.

This simulation illustrates how the structure of the environment is acquired separately from the location of resources. The agent can explore and learn the map of the environment even without any resources present (*Figure 4B*). This learning takes place among the map synapses in the endotaxis circuit (*Figure 1B*). When a resource is found, its location gets tagged within that established map through learning by the goal synapses. The resulting goal signal is available immediately without the need for further learning (*Figure 4C*). If the distribution of resources changes, the knowledge in the map remains unaffected (*Figure 4D*) but the goal synapses can change quickly to incorporate the new target. Vice versa, if the graph of the environment changes, the map synapses get updated, and that adapts the goal signal to the new situation even without further change in the goal synapses (*Figure 1E*).

What happens if a previously existing link disappears from the environment, for example, because one corridor of the mouse burrow caves in? Ideally the agent would erase that link from the cognitive map. The learning algorithm (Algorithm 2) is designed for rapid and robust acquisition of a cognitive map starting from zero knowledge and does not contain a provision for forgetting. However, one can add a biologically plausible rule for synaptic depression that gradually erases memory of a link if the agent never travels it. Details are presented in section 'Forgetting of links and resources' (Figure 10). For the sake of simplicity, we continue the present analysis of endotaxis based on the simple three-parameter algorithm presented above (Algorithm 2).

## Choice of learning rule

The map learning rule in Algorithm 2 produces full-strength synapses $M_{ij}$ and $M_{ji}$ after a single co-activation of the two neurons. A more common approach to synaptic learning uses small incremental updates and stabilizes the update rule with some form of normalization, based on the average pre- or postsynaptic activity over many steps (*Gerstner and Kistler, 2002*). For example, presynaptic normalization leads the synaptic network to learn a transition probability matrix (*Fang et al., 2023*)

$$T_{ij} = \text{probability of stepping to node } i \text{ given current node } j$$

Instead, we adopted the instantaneous update model for two reasons: most importantly, this allows the agent to learn a route after the first traversal, which is needed to explain the rapid learning observed in experimental animals. For example, section 'Navigating a partial map: homing behavior' models accurate homing after the first excursion into the labyrinth. Furthermore, when we repeated the analysis of *Figure 3* using the transition matrix $T_{ij}$ instead of the adjacency matrix $A_{ij}$, the goal signal correlated more weakly with distance, and even with the optimal gain setting the range of correct navigation was considerably reduced.

This rapid learning rule reflects an implicit assumption that the environment is static, such that the learned transition will always be available. For adaptation to slow changes in the environment, see section 'Forgetting of links and resources.' Note also that the above procedure Algorithm 2 updates both synapses between neurons $i$ and $j$. This assumes implicitly that the experienced edge on the graph can also be traversed in the opposite direction, which applies to many navigation problems. To learn a directed environment – such as a city map with one-way streets or a game in which moves cannot be reversed – one may use a directed learning rule that requires the presynaptic neuron to fire before the postsynaptic neuron. This will update only the synapse $M_{ij}$ representing the edge that was actually traveled. For all simulations in this article, we will use the symmetric learning rule.

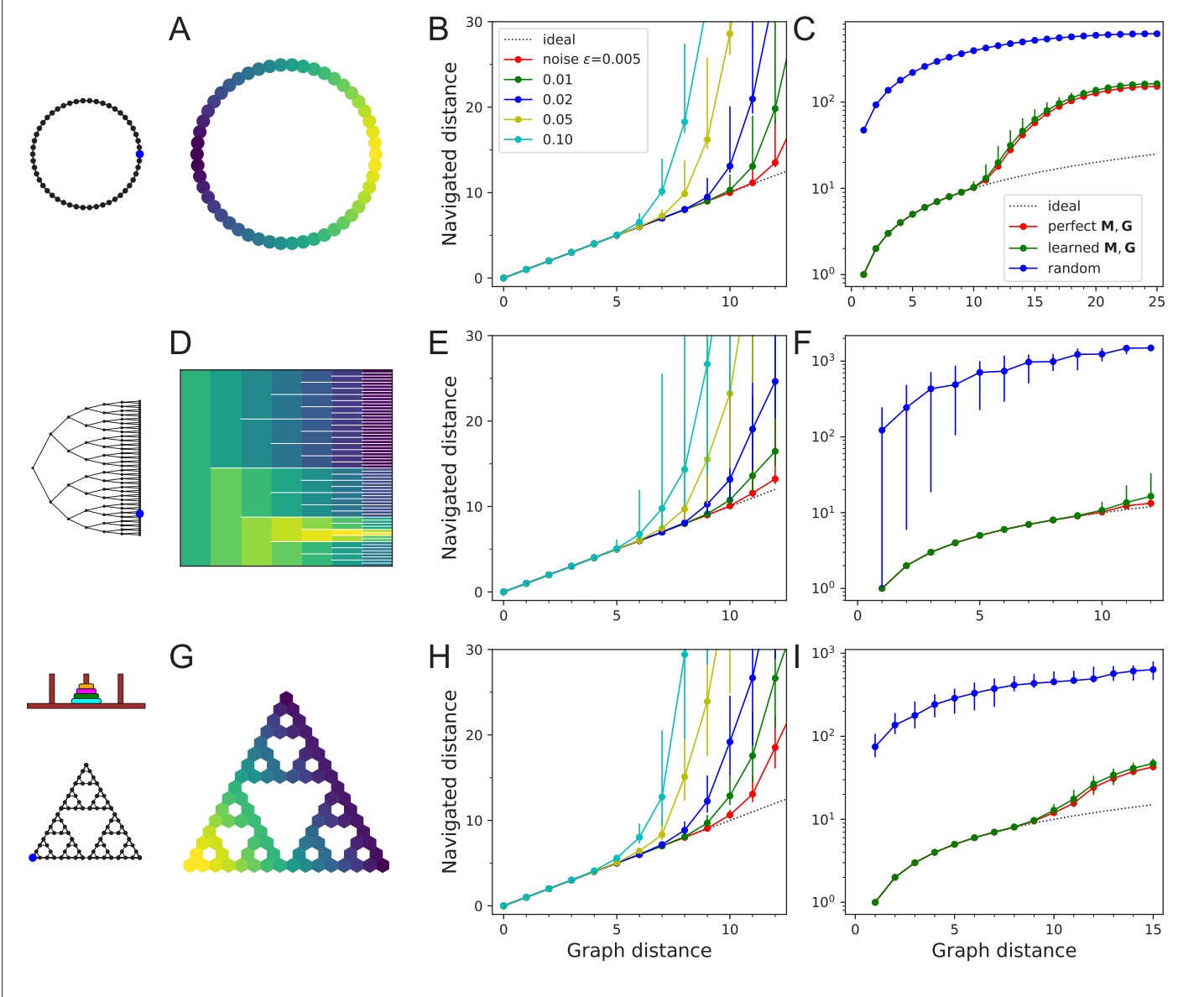

**Figure 5.** Navigation using the learned map and targets. (**A–C**) Ring with 50 nodes. (**A**) Goal signal for a single target location (blue dot on left icon) after learning during random exploration with 10,000 steps. Color scale is logarithmic, yellow = high. Note the monotonic decay of the goal signal with graph distance from the target. (**B**) Results of all-to-all navigation where every node is a separate goal. For all pairs of nodes, this shows the navigated distance vs the graph distance. Median ±10/90 percentiles for all routes with the same graph distance. 'Ideal' navigation would follow the identity. The actual navigation is ideal over short distances, then begins to deviate from ideal at a critical distance that depends on the noise level $\epsilon$. (**C**) As in (**B**) over a wider range, note logarithmic axis. Noise $\epsilon = 0.01$. Includes comparison to navigation by a random walk; and navigation using the optimal goal signal based on knowledge of the graph structure and target location. $\gamma = 0.41, \theta = 0.39, \alpha = 0.1$. (**D–F**) As in (**A–C**) for a binary tree graph with 127 nodes. (**D**) Goal signal to the node marked on the left icon. This was the reward port in the labyrinth experiments of **Rosenberg et al., 2021**. White lines separate the branches of the tree. $\gamma = 0.33, \theta = 0.30, \alpha = 0.1$. (**G–I**) As in (**A–C**) for a 'Tower of Hanoi' graph with 81 nodes. $\gamma = 0.29, \theta = 0.27, \alpha = 0.1$.

## Navigation using the learned goal signal

We now turn to the 'exploitation' component of endotaxis, namely use of the learned information to navigate toward targets. In the simulations of *Figure 5*, we allow the agent to explore a graph. Every node on the graph drives a separate resource cell, thus the agent simultaneously learns goal signals to every node. After a random walk sufficient to cover the graph several times, we test the agent's ability to navigate to the goals by ascending on the learned goal signal. For that purpose, we teleport the

agent to an arbitrary start node in the graph and ask how many steps it takes to reach the goal node following the policy of Algorithm 1. In these tests, the learning of map and goal synapses was turned off during the navigation phase, so we could separately assess how learning and navigating affect the performance. However, there is no functional requirement for this, and indeed one of the attractive features of this model is that learning and navigation can proceed in parallel at all times.

*Figure 5A–C* shows results on a ring graph with 50 nodes. With suitable values of the model parameters $(\gamma, \theta, \alpha)$ – more on that later – the agent learns a goal signal that declines monotonically with distance from the target node (*Figure 5A*). The ability to ascend on that goal signal depends on the noise level $\epsilon$, which determines whether the agent can sense the difference in goal signal at neighboring nodes. At a high noise level $\epsilon = 0.1$, the agent finds the target by the shortest route from up to five links away (*Figure 5B*); beyond that range, some navigation errors creep in. At a low noise level of $\epsilon = 0.005$, navigation is perfect up to 10 links away. Every factor of two increase in noise seems to reduce the range of navigation by about one link.

How does the process of learning the map of the environment affect the ultimate navigation performance? *Figure 5C* makes that comparison by considering an agent with oracular knowledge of the graph structure and target location (*Equations 9* and *10*). Interestingly, this barely improves the distance range for perfect navigation. By contrast, an agent performing a random walk with zero knowledge of the environment would take about 40 times longer to reach the target than by using endotaxis (*Figure 5C*).

The ring graph is particularly simple, but how well does endotaxis learn in a more realistic environment? *Figure 5D–F* shows results on a binary tree graph with six levels: this is the structure of a maze used in a recent study on mouse navigation (*Rosenberg et al., 2021*). In those experiments, mice learned quickly how to reach the reward location (blue dot in *Figure 5D*) from anywhere within the maze. Indeed, the endotaxis agent can learn a goal signal that declines monotonically with distance from the reward port (*Figure 5D*). At a noise level of $\epsilon = 0.01$, navigation is perfect over distances of 9 links and close to perfect over the maximal distance of 12 links that occurs in this maze (*Figure 5E*). Again, the challenge of having to learn the map affects the performance only slightly (*Figure 5F*). Finally, comparison with the random agent shows that endotaxis shortens the time to target by a factor of 100 on this graph (*Figure 5F*).

*Figure 5G–I* shows results for a more complex graph that represents a cognitive task, namely the game 'Tower of Hanoi.' Disks of different sizes are stacked on three pegs, with the constraint that no disk can rest on top a smaller one. The game is solved by rearranging the pile of disks from the center peg to another. In any state of the game, there are either two or three possible actions, and they form an interesting graph with many loops (*Figure 5G*). The player starts at the top node (all disks on the center peg) and the two possible solutions correspond to the bottom left and right corners. Again, random exploration leads the endotaxis agent to learn the connectivity of the game and to discover the solutions. The resulting goal signal decays systematically with graph distance from the solution (*Figure 5G*). At a noise of $\epsilon = 0.01$, navigation is perfect once the agent gets to within nine moves of the target (*Figure 5H*). This is not quite sufficient for an error-free solution from the starting position, which requires 15 moves. However, compared to an agent executing random moves, endotaxis speeds up the solution by a factor of 10 (*Figure 5I*). If the game is played with only three disks, the maximal graph distance is 7, and endotaxis solves it perfectly at $\epsilon = 0.01$.

These results show that endotaxis functions well in environments with very different structure: linear, tree-shaped, and cyclic. Random exploration in conjunction with synaptic learning can efficiently acquire the connectivity of the environment and the location of targets. With a noise level of 1%, the resulting goal signal allows perfect navigation over distances of ~9 steps, independent of the nature of the graph. This is a respectable range: personal experience suggests that we rarely learn routes that involve more than nine successive decisions. Chess openings, which are often played in a fast and reflexive fashion, last about 10 moves.

## Parameter sensitivity

The endotaxis model has only three parameters: the gain $\gamma$ of map units, the threshold $\theta$ for learning at map synapses, and the learning rate $\alpha$ at goal synapses. How does performance depend on these parameters? Do they need to be tuned precisely? And does the optimal tuning depend on the spatial environment? There is a natural hierarchy to the parameters if one separates the process of learning

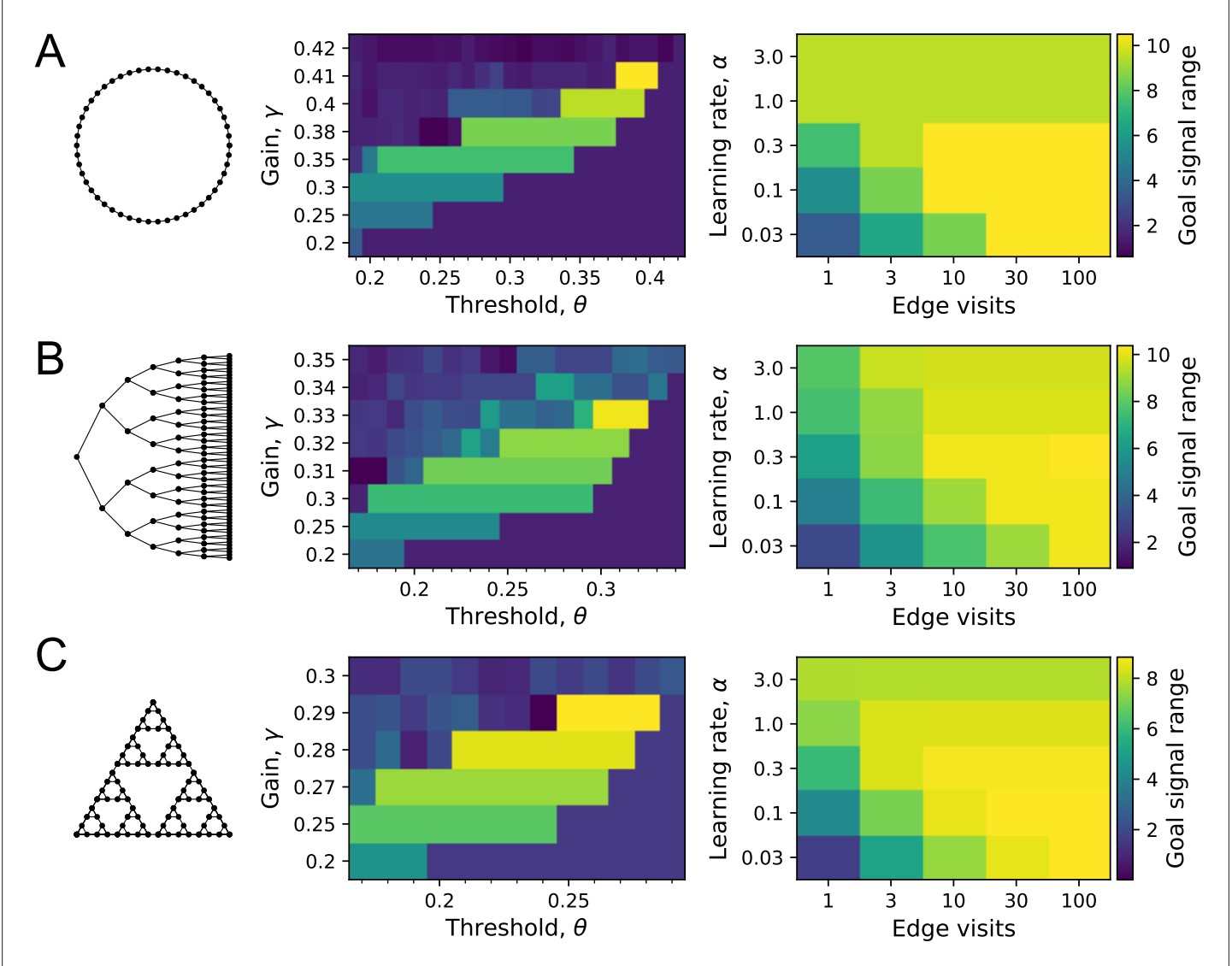

**Figure 6.** Sensitivity of performance to the model parameters. On each of the three graphs, we simulated endotaxis for all-to-all navigation, where each node serves as a start and a goal node. The performance measure was the range of the goal signal, defined as the graph distance over which at least half the navigated routes follow the shortest path. The exploration path for synaptic learning was of medium length, visiting each edge on the graph approximately 10 times. The noise was set to $\epsilon = 0.01$. (**A**) Ring graph with 50 nodes. Left: dependence of the goal signal range on the gain $\gamma$ and the threshold $\theta$ for learning map synapses. Performance increases with higher gain until it collapses beyond the critical value. For each gain, there is a sharply defined range of useful thresholds, with lower values at lower gain. Right: dependence of the goal signal range on the learning rate $\alpha$ at goal synapses, and the length of the exploratory walk, measured in visits per edge of the graph. For a short walk (one edge visit), a high learning rate is best. For a long walk (100 edge visits), a lower learning rate wins out. (**B**) As in (**A**) for the Binary tree maze with 127 nodes. (**C**) As in (**A**) for the Tower of Hanoi graph with 81 nodes.

from that of navigation. Suppose the circuit has learned the structure of the environment perfectly, such that the map synapses reflect the adjacencies (*Equation 9*), and the goal synapses reflect the map output at the goal (*Equation 10*). Then the optimal navigation performance of the endotaxis system depends only on the gain $\gamma$ and the noise level $\epsilon$. For a given $\gamma$, in turn, the precision of map learning depends only on the threshold $\theta$ (see Algorithm 2). Finally, if the gain is set optimally and the map was learned properly, the identification of targets depends only on the goal-learning rate $\alpha$. *Figure 6* explores these relationships in turn.

We simulated the learning phase of endotaxis as in the preceding section (*Figure 5B, E and H*), using a noise level of $\epsilon = 0.01$, and systematically varying the model parameters ($\gamma, \theta, \alpha$). For each

parameter set, we measured the graph distance over which at least half of the navigated routes were perfect. We defined this distance as the range of the goal signal.

For example, on the ring graph (*Figure 6A*) the signal range improves with gain until performance collapses beyond a maximal gain value. This is just as predicted by the theory (*Figure 3*), except that the maximal gain $\gamma_{max} = 0.41$ is somewhat below the critical value $\gamma_c = 0.5$. Clearly the added complications of having to learn the map and goal locations take their toll at high gain. Below the maximal cutoff, the dependence of performance on gain is rather gentle: for example, a 14% change in gain from 0.35 to 0.40 leads to a 26% change in performance. At any given gain value, there is a range of values for the threshold $\theta$ within which the map is learned perfectly. Note that this range is generous and does not require precise adjustment: for example, under a near-maximal gain of 0.38, the threshold can vary freely over a 35% range.

Once the gain and synaptic threshold are set so as to acquire the map synapses, the quality of goal-learning depends only on the learning rate $\alpha$. With large $\alpha$, a single visit to the goal fully potentiates the goal synapses so that they do not get updated further. This allows for a fast acquisition of that target, but at the risk of imperfect learning, because the map may not be fully explored yet. A small $\alpha$ will update the synapses only partially over many successive visits to the goal. This leads to a poor performance after short exploration, because the weak goal signal competes with noise, but superior performance after long explorations: a tradeoff between speed of learning and accuracy. Precisely this speed-accuracy tradeoff is seen in the simulations (*Figure 6A*, right): a high learning rate is optimal for short explorations, but for longer ones a small learning rate wins out. An intermediate value of $\alpha = 1$ delivers a good compromise performance.

We found qualitatively similar behavior for the other two environments studied here: the binary maze graph (*Figure 6B*) and the Tower of Hanoi graph (*Figure 6C*). In each case, the maximal usable

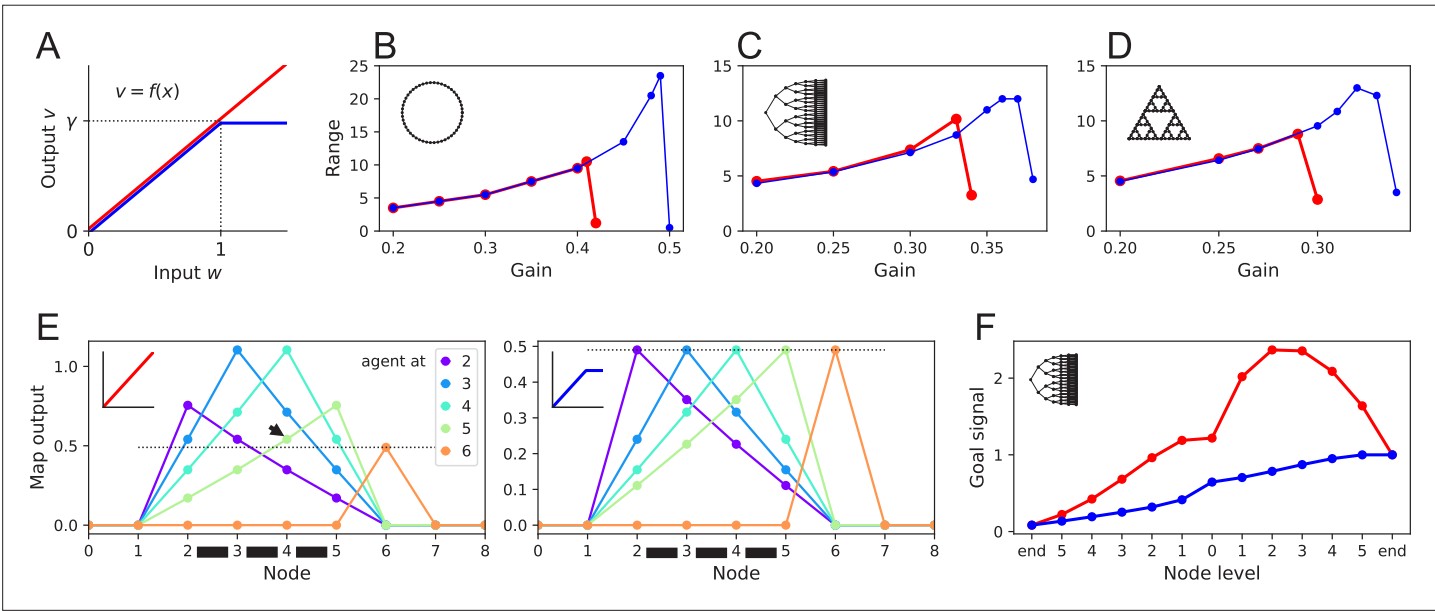

**Figure 7.** Benefits of a nonlinear activation function. (**A**) The activation function relating a map neuron's output $v$ to its total input $w$. Red: linear function with gain $\gamma$. Blue: nonlinear function with saturation at $w > 1$. (**B–D**) Range of the goal signal, as defined in *Figure 6*, as a function of the gain $\gamma$ (noise $\epsilon = 0.01$). Range increases with gain up to a maximal value. The maximal range achieved is higher with nonlinear activation (blue) than linear activation (red). Results for the ring graph (**B**), binary tree maze (**C**), and Tower of Hanoi graph (**D**). (**E**) Output of map cells during early exploration of the ring graph (gain $\gamma = 0.49$). Suppose the agent has walked back and forth between nodes 2 and 5, so all their corresponding map synapses are established (black bars). Then the agent steps to node 6 for the first time (orange). Lines plot the output of the map cells with the agent at locations 2, 3, 4, 5, or 6. Dotted line indicates the maximal possible setting of the threshold $\theta$ in the learning rule. With linear activation (left), a map cell receiving purely recurrent input (4) may produce a signal larger than threshold (arrowhead above the dotted line). Thus, cells 4 and 6 would form an erroneous synapse. With a saturating activation function (right), the map amplitude stays constant throughout learning, and this confound does not happen. (**F**) The goal signal from an end node of the binary maze, plotted along the path from another end node. Map and goal synapses set to their optimal values assuming full knowledge of the graph and the target (gain $\gamma = 0.37$). With linear activation (red), the goal signal has a local maximum, so navigation to the target fails. With a saturating activation function (blue), the goal signal is monotonic and leads the agent to the target.

gain is slightly below the critical value $\gamma_c$ of that graph. A learning rate of $\alpha = 1$ delivers intermediate results. For long explorations, a lower learning rate is best.

In summary, this sensitivity analysis shows that the optimal parameter set for endotaxis does depend on the environment. This is not altogether surprising: every neural network needs to adapt to the distribution of inputs it receives so as to perform optimally. At the same time, the required tuning is rather generous, allowing at least 10–20% slop in the parameters for reasonable performance. Furthermore, a single parameter set of $\gamma = 0.29, \theta = 0.26, \alpha = 1$ performs quite well on both the binary maze and the Tower of Hanoi graphs, which are dramatically different in character.

## A saturating activation function improves navigation

So far, the model of the map network used neurons with a linear activation function (*Equation 3*), meaning the output $v$ is simply proportional to the input, $v = \gamma w$. We also explored nonlinear activation functions $v = f(w)$ and found that the performance of endotaxis improves under certain conditions (*Fang et al., 2023*). The most important feature is that $f(w)$ should saturate for inputs $x$ that are larger than the output of the point cells ($u = 1$ in *Equation 4*). The detailed shape matters little, so for illustration we will use a linear-flat activation curve (*Figure 7A*):

$$f(w) = \begin{cases} \gamma w, & \text{if } w \leq 1 \\ \gamma, & \text{if } w > 1 \end{cases} \tag{19}$$

*Figure 7B–D* reports the range of navigation on the three sample graphs, defined and computed from simulations as in the preceding section (*Figure 6*). The effective range is the largest graph distance over which the median trajectory chooses the shortest route. As observed using linear map neurons (*Figure 6*), the range increases with the gain $\gamma$ until it collapses beyond some maximal value (*Figure 7B–D*). However, the saturating activation function allowed for higher gain values, which led to considerable increases in the range of navigation: by a factor of 2.2 for the ring graph, and 1.5 for the Tower of Hanoi graph. On the binary maze, the saturating activation function allowed perfect navigation over the maximal distance available of 12 steps.

The enhanced performance was a result of better map learning as well as better navigation. To understand the former, consider *Figure 7E*: here the agent has begun to learn the ring graph by walking back and forth between a few nodes (2–5), thus establishing all their pairwise map synapses; then it steps to a new node (6). With a linear activation function (*Figure 7E*, left), the recurrent synapses enhance the map output, so the map signal with the agent in the explored region (2–5) is considerably larger than after stepping to the new node. This interferes with the mechanism for map learning: the learning rule must identify which of the map cells represents the current location of the agent, and does so by setting a threshold on the output signal (Algorithm 2). In the present example, this leads to erroneous synapses because a map cell that receives only recurrent input (4) produces outputs larger than the threshold (arrowhead in *Figure 7E*). With the saturating activation function (*Figure 7E*, right), the directly activated map cells always have the largest output signal, so the learning rule can operate without errors.

The saturating activation function also helps after learning is complete. In *Figure 7F*, the agent is given perfect knowledge of the binary maze map, then asked to use the resulting goal signals to navigate from one end node to another. With a linear activation function, the goal signal has a large local maximum that traps the agent. The nonlinear activation function produces a monotonic goal signal that leads the agent to the target.

Both these aspects of enhanced performance can be traced to the normalizing effect of the nonlinearity that keeps the peak output of the map constant. Such normalization could be performed by other mechanisms as well, for example, a global inhibitory feedback among the map neurons.

In summary, this section shows that altering details of the model can substantially extend its performance. For the remainder of this article, we will return to the linear activation curve because interesting behavioral phenomena can be observed even with the simple linear model.

## Navigating a partial map: Homing behavior

We have seen that endotaxis can learn both connections in the environment and the locations of targets after just one visit (*Figure 6*.) This suggests that the agent can navigate well on whatever

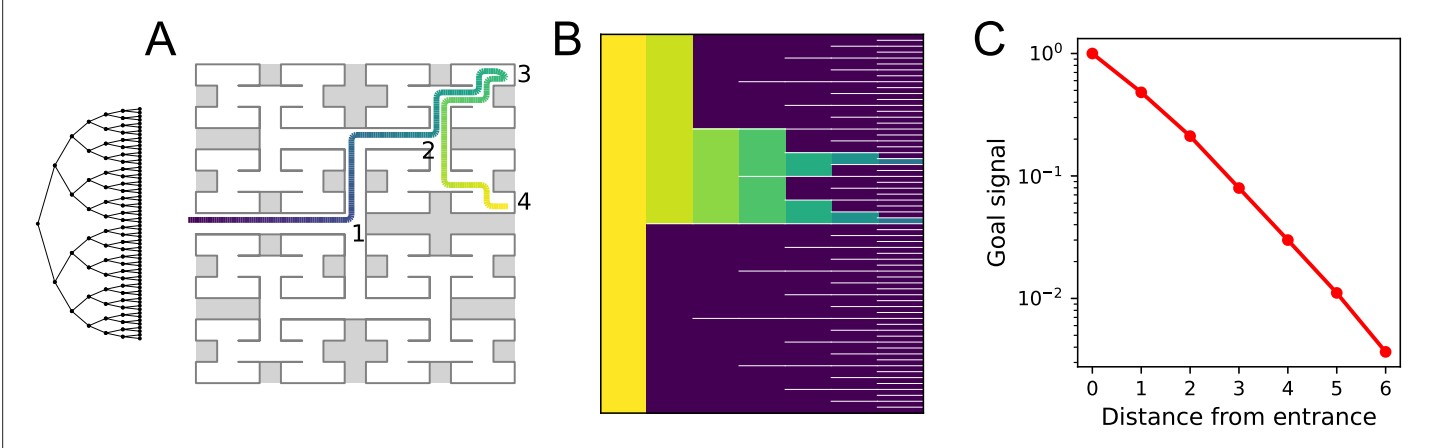

**Figure 8.** Homing by endotaxis. (**A**) A binary tree maze as used in *Rosenberg et al., 2021*. A simulated mouse begins to explore the labyrinth (colored trajectory, purple = early, yellow = late), traveling from the entrance (1) to one of the end nodes (3), then to another end node (4). Can it return to the entrance from there using endotaxis? (**B**) Goal signal learned by the end of the walk in (**A**), displayed as in *Figure 5D*, purple = 0. Note the goal signal is nonzero only at the nodes that have been encountered so far. From all those nodes, it increases monotonically toward the entrance. (**C**) Detailed plot of the goal signal along the shortest route for homing. Parameters $\gamma = 0.33, \theta = 0.30, \alpha = 10, \epsilon = 0.01$.

portion of the environment it has already seen, before covering it exhaustively. To illustrate this, we analyze an ethologically relevant instance.

Consider a mouse that enters an unfamiliar environment for the first time, such as a labyrinth constructed by graduate students (*Rosenberg et al., 2021*). Given the uncertainties about what lurks inside, the mouse needs to retain the ability to flee back to the entrance as fast as possible. For concreteness, take the mouse trajectory in *Figure 8A*. The animal has entered the labyrinth (location 1), made its way to one of the end nodes (3), then explored further to another end node (4). Suppose it needs to return to the entrance now. One way would be to retrace all its steps. But the shorter way is to take a left at (2) and cut out the unnecessary branch to (3). Experimentally we found that mice indeed take the short direct route instead of retracing their path (*Rosenberg et al., 2021*). They can do so even on the very first visit of an unfamiliar labyrinth. Can endotaxis explain this behavior?

We assume that the entrance is a salient location, so the agent dedicates a goal cell to the root node of the binary tree. *Figure 8B* plots the goal signal after the path in panel A, just as the agent wants to return home. The goal signal is nonzero only at the locations the agent has visited along its path. It clearly increases monotonically toward the entrance (*Figure 8C*). At a noise level of $\epsilon = 0.01$, the agent can navigate to the entrance by the shortest path without error. Note specifically that the agent does not retrace its steps when arriving at location (2), but instead turns toward (1).

One unusual aspect of homing is that the goal is identified first, before the agent has even entered the environment to explore it. That strengthens the goal synapse from the sole map cell that is active at the entrance. Only subsequently does the agent build up map synapses that allow the goal signal to spread throughout the map network. Still, in this situation, the single synapse onto the goal cell is sufficient to convey a robust signal for homing.

## Efficient patrolling

Beside exploring and exploiting, a third mode of navigating the environment is patrolling. At this stage, the animal knows the lay of the land, and has perhaps discovered some special locations, but continues to patrol the environment for new opportunities or threats. In our study of mice freely interacting with a large labyrinth, the animals spent more than 85% of the time patrolling the maze (*Rosenberg et al., 2021*). This continued for hours after they had perfected the targeting of reward locations and the homing back to the entrance. Presumably, the goal of patrolling is to cover the entire environment frequently and efficiently so as to spot any changes as soon as they develop. So the ideal path in patrolling would visit every node on the graph in the smallest number of steps possible. In the binary tree maze used for our experiments, that optimal patrol path takes 252 steps: it visits every end node of the labyrinth exactly once without any repeats (*Figure 9A*).

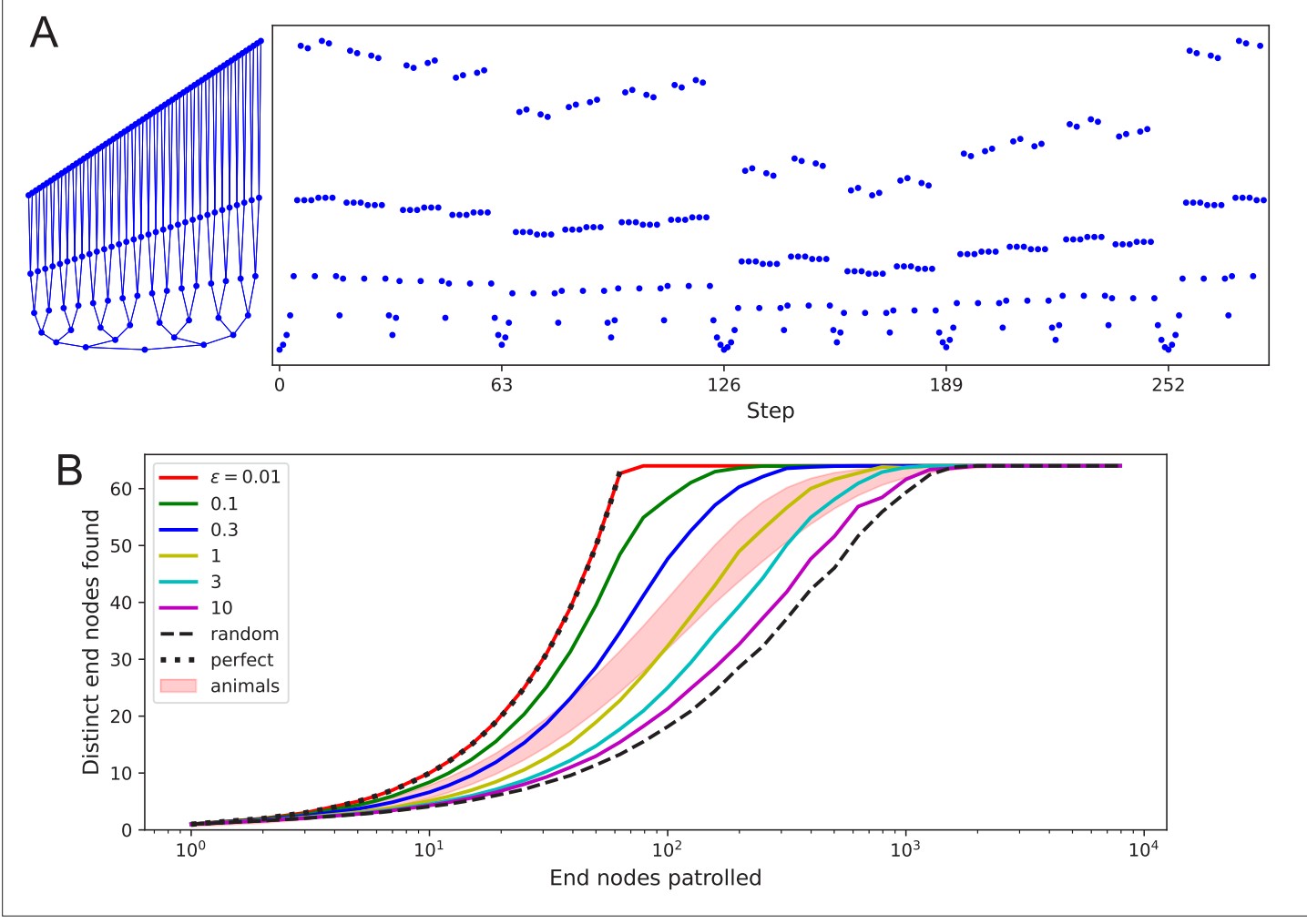

**Figure 9.** Patrolling by endotaxis. (**A**) Left: a binary tree maze as used in *Rosenberg et al., 2021*, plotted here so every node has a different vertical offset. Right: a perfect patrol path through this environment. It visits every node in 252 steps, then starts over. (**B**) Patrolling efficiency of different agents on the binary tree maze. The focus here is on the 64 end nodes of the labyrinth. We ask how many distinct end nodes are found (vertical axis) as a function of the number of end nodes visited (horizontal axis). For the perfect patrolling path, that relationship is the identity ('perfect'). For a random walk, the curve is shifted far to the right ('random', note log axis). Ten mice in *Rosenberg et al., 2021* showed patrolling behavior within the shaded range. Solid lines are the endotaxis agent, operating at different noise levels $\epsilon$. Note $\epsilon = 0.01$ produces perfect patrolling; in fact, panel (**A**) is a path produced by this agent. Higher noise levels lead to lower efficiency. The behavior of mice corresponds to $\epsilon \approx 1$. Gain $\gamma = 0.33$, habituation $\beta = 1.2$, with recovery time $\tau = 100$ steps.

Real mice do not quite execute this optimal path, but their patrolling behavior is much more efficient than random (*Figure 9B*). They avoid revisiting areas they have seen recently. Could endotaxis implement such an efficient patrol of the environment? The task is to steer the agent to locations that have not been visited recently. One can formalize this by imagining a resource called 'neglect' distributed throughout the environment. At each location, neglect increases with time, then resets to zero the moment the agent visits there. To use this in endotaxis, one needs a goal cell that represents neglect.

We add to the core model a goal cell that represents 'neglect.' It receives excitation from every map cell via synapses that are equal and constant in strength (see clock symbol in *Figure 1B*). This produces a goal signal that is approximately constant everywhere in the environment. Now suppose that the point neurons undergo a form of habituation: when a point cell fires because the agent walks through its field, its sensitivity decreases by some habituation factor. That habituation then decays over time until the point cell recovers its original sensitivity. As a result, the most recently visited points

on the graph produce a smaller goal signal. Endotaxis based on this goal signal will therefore lead the agent to the areas most in need of a visit.

*Figure 9B* illustrates that this is a powerful way to implement efficient patrols. Here we modeled endotaxis on the binary tree labyrinth, using the standard parameters useful for exploration, exploitation, and homing in previous sections. To this, we added a habituation in the point cells with exponential recovery dynamics. Formally, the procedure is defined by Algorithm 3. Again, we turned off the learning rules (Algorithm 2) during this simulation to observe the effects of habituation in isolation. A fully functioning agent can keep the learning rules on at all times (Figure 11).

---

**Algorithm 3 Patrolling.**

---

Parameters: gain $\gamma$, noise $\epsilon$, habituation $\beta$, recovery time $\tau$
Input: map synapses $\mathbf{M}$

$\quad h_i \leftarrow 1$, for all point cells $i$       ▷ starting sensitivity of point cell at node $i$

$\quad s \leftarrow x$       ▷ begin patrolling at node $x$

$\quad$ **while** patrolling **do**

$\quad\quad h_s \leftarrow h_s\, e^{-\beta}$       ▷ habituation of point cell $s$

$\quad\quad h_i \leftarrow 1 - (1 - h_i)\, e^{-1/\tau}$, for all $i$       ▷ resensitization of all point cells

$\quad\quad$ **for** all nodes $j$ that neighbor $s$ **do**       ▷ agent tests available options

$\quad\quad\quad u_i(j) \leftarrow \delta_{i,j} h_j$ for all $i$       ▷ point cell output with agent at node $j$

$$\mathbf{v}(j) \leftarrow \left(\frac{1}{\gamma}\mathbf{1} - \mathbf{M}\right)^{-1} \mathbf{u}(j)$$       ▷ map output

$$p(j) \leftarrow \frac{1}{Z}\sum_i v_i(j) + \eta$$       ▷ sum of map output with noise, normalized so max = 1

$\quad\quad$ **end for**

$\quad\quad s \leftarrow \arg\max_j p(j)$       ▷ move to neighbor node with the highest patrol signal

$\quad$ **end while**

---

With appropriate choices of habituation $\beta$ and recovery time $\tau$, the agent does in fact execute a perfect patrol path on the binary tree, traversing every edge of the graph exactly once, and then repeating that sequence indefinitely (*Figure 9A*). For this to work, some habituation must persist for the time taken to traverse the entire tree; in this simulation, we used $\tau = 100$ steps on a graph that requires 252 steps. As in all applications of endotaxis, the performance also depends on the readout noise $\epsilon$. For increasing readout noise, the agent's behavior transitions gradually from the perfect patrol to a random walk (*Figure 9B*). The patrolling behavior of real mice is situated about halfway along that range, at an equivalent readout noise of $\epsilon = 1$ (*Figure 9B*).

Finally, this suggests a unified explanation for exploration and patrolling: in both modes, the agent follows the output of the 'neglect' cell, which is just the sum total of the map output. However, in the early exploration phase, when the agent is still assembling the cognitive map, it gives the neglect signal zero or low weight, so the turning decisions are dominated by the readout noise and produce something close to a random walk. Later on, the agent assigns a higher weight to the neglect signal, so it exceeds the readout noise and shifts the behavior toward systematic patrolling. In our simulations, an intrinsic readout noise of $\epsilon = 0.01$ is sufficiently low to enable even a perfect patrol path (*Figure 9B*).

In summary, the core model of endotaxis can be enhanced by adding a basic form of habituation at the input neurons. This allows the agent to implement an effective patrolling policy that steers towards regions which have been neglected for a while. Of course, habituation among point cells will also change the dynamics of map learning during the exploration phase. We found that both map and goal synapses are still learned effectively, and navigation to targets is only minimally affected by habituation (Figure 11).

## Discussion

### Summary of claims

We have presented a biologically plausible neural mechanism that can support learning, navigation, and problem solving in complex environments. The algorithm, called *endotaxis*, offers an end-to-end

solution for assembling a cognitive map (*Figure 4*), memorizing interesting targets within that map, navigating to those targets (*Figure 5*), as well as accessory functions like instant homing (*Figure 8*) and effective patrolling (*Figure 9*). Conceptually, it is related to chemotaxis, namely the ability to follow an odor signal to its source, which is shared universally by most or all motile animals. The endotaxis network creates an internal 'virtual odor' which the animal can follow to reach any chosen target location (*Figure 1*). When the agent begins to explore the environment, the network learns both the structure of the space, namely which points are connected, and the location of valuable resources (*Figure 4*), even after a single experience (*Figures 4 and 8*). The agent can then navigate back to those target locations efficiently from any point in the environment (*Figure 5*). Beyond spatial navigation, endotaxis can also learn the solution to purely cognitive tasks (*Figure 5*) that can be formulated as search on a graph (section 'Theory of endotaxis'). It takes as given two elementary facts: the existence of place cells that fire when the animal is at a specific location, and a behavioral module that allows the animal to follow an odor gradient uphill. The proposed circuit (*Figure 1*) provides the interface from the place cells to the virtual odor gradient. In the following sections, we consider how these findings relate to phenomena of animal behavior and neural circuitry, and prior art in the area of theory and modeling.

## Theories and models of spatial learning

Broadly speaking, endotaxis can be seen as a form of reinforcement learning (*Sutton and Barto, 2018*): the agent learns from rewards or punishments in the environment and develops a policy that allows for subsequent navigation to special locations. The goal signal in endotaxis plays the role of a value function in reinforcement learning theory. From experience, the agent learns to compute that value function for every location and control its actions accordingly. Within the broad universe of reinforcement learning algorithms, endotaxis combines some special features as well as limitations that are inspired by empirical phenomena of animal learning, and also make it suitable for a biological implementation.

First, most of the learning happens without any reinforcement. During the exploratory random walk, endotaxis learns the topology of the environment, specifically by updating the synapses in the map network ($\mathbf{M}$ in *Figure 1B*). Rewards are not needed for this map learning, and indeed the goal signal remains zero during this period (*Figure 4*). Once a reward is encountered, the goal synapses ($\mathbf{G}$ in *Figure 1B*) get set, and the goal signal instantly spreads through the known portion of the environment. Thus, the agent learns how to navigate to the goal location from a single reinforcement (*Figure 4*). This is possible because the ground has been prepared, as it were, by learning a map. In animal behavior, the acquisition of a cognitive map without rewards is called *latent learning*. Early debates in animal psychology pitched latent learning and reinforcement learning as alternative explanations (*Thistlethwaite, 1951*). Instead, in the endotaxis algorithm, neither can function without the other as the goal signal explicitly depends on both the map and goal synapses (*Equation 18*, Algorithm 1).

More specifically, the neural signals in endotaxis bear some similarity to the so-called *successor representation* (*Dayan, 1993*; *Corneil and Gerstner, 2015*; *Stachenfeld et al., 2017*; *Garvert et al., 2017*; *Fang et al., 2023*). This is a proposal for how the brain might encode the current state of the agent, intended to simplify the mathematics of time-difference reinforcement learning. In that representation, there is a neuron for every state of the agent, and the activity of neuron $j$ is the time-discounted probability that the agent will find itself at state $j$ in the future. Similarly, the output of the endotaxis map network is related to future states of the agent and follows a similar functional dependence on distance (*Meister, 2023*, *Equation 7*). However, despite these formal similarities, the underlying logic is quite different. In the successor representation, $\gamma$ plays the role of a temporal discount factor for rewards (*Dayan, 1993*); essentially it is the proportionality factor in the agent's belief that 'time is money.' In this picture, varying $\gamma$ allows the agent to make predictions with different time horizons (*Fang et al., 2023*; *Stachenfeld et al., 2017*). In endotaxis, there is no time/reward tradeoff. The agent simply wants the shortest path to the goal. The map network reflects the objective connectivity of the environment to the farthest extent possible. Here $\gamma$ is the gain of the map neurons that, when properly chosen, allows the neural network to perform that computation. The agent may want to tune $\gamma$ to the statistics of the environment, although we showed that a common value of $\gamma$ works quite well across environments (*Figure 6*). (These differences in how the problem is formulated can lead to slightly different mathematical expressions, for example, compare the role of $\gamma$ in *Equation 7* with Equation 2 of *Fang et al., 2023*.)

Second, endotaxis does not tabulate the list of available actions at each state. That information remains externalized in the environment: the agent simply tries whatever actions are available at the moment, then picks the best one. This is a characteristically biological mode of action and most organisms have a behavioral routine that executes such trial-and-error. This 'externalized cognition' simplifies the learning task: for any given navigation policy, the agent needs to learn only one scalar function of location, namely the goal signal. By comparison, many machine learning algorithms develop a value function for state–action pairs, which then allows more sophisticated planning (*Sutton and Barto, 2018*; *Moerland et al., 2023*). The relative simplicity of the endotaxis circuit depends on the limitation to learning only state functions.

Some key elements of the endotaxis model have appeared in prior work, starting with the notion of ascending a scalar goal signal during navigation (*Schmajuk and Thieme, 1992*; *Voicu and Schmajuk, 2000*; *Samsonovich and Ascoli, 2005*). Several models assume the existence of a map layer, in which individual neurons stand for specific places, and the excitatory synapses between neurons represent the connections between those places (*Gaussier et al., 2002*; *Schölkopf and Mallot, 1995*; *Voicu and Schmajuk, 2000*; *Trullier and Meyer, 2000*; *Martinet et al., 2011*; *Ponulak and Hopfield, 2013*; *Khajeh-Alijani et al., 2015*). Then the agent somehow reads out those connections in order to find the shortest path between its current location (the start node) and a desired target (the end node).

Very different schemes have been proposed for this readout of the map. The most popular scheme is to somehow inject a signal into the desired end node, let it propagate backward through the network, and read out the magnitude or gradient of the signal near the start node (*Glasius et al., 1996*; *Gaussier et al., 2002*; *Gorchetchnikov and Hasselmo, 2005*; *Martinet et al., 2011*; *Ponulak and Hopfield, 2013*; *Khajeh-Alijani et al., 2015*). In general, this requires some accessory system that can look up which neuron in the map corresponds to the desired end node, and which neuron to the agent's current location or its neighbors; often these accessory functions remain unspecified (*Schölkopf and Mallot, 1995*; *Voicu and Schmajuk, 2000*; *Khajeh-Alijani et al., 2015*). By contrast, in the endotaxis model the signal is propagated in the forward direction starting with the activity of the place cell at the agent's current location. The signal strength is read out at the goal location: The goal neuron is the same neuron that also responds directly to the rewarding feature at the goal location. For example, the proximity to water is read out by a neuron that is also excited when the animal drinks water. In this way, the brain does not need to maintain a separate lookup table for goal neurons. If the agent wants to find water, it should simply follow the same neuron that fires when it drinks.

Another distinguishing feature of endotaxis is that it operates continuously. Many models for navigation have to separate the phase of spatial learning from the phase of goal-directed navigation. Sometimes plasticity needs to be turned off or reset during one phase or the other (*Samsonovich and Ascoli, 2005*; *Ponulak and Hopfield, 2013*). Sometimes a special signal must be injected during goal-seeking (*Voicu and Schmajuk, 2000*). Sometimes the rules change depending on whether the agent approaches or leaves a target (*Blum and Abbott, 1996*). Again this requires additional supervisory systems that often go unexplained. By contrast, endotaxis is 'always on.' Whether the animal explores a new environment, navigates to a target, or patrols a well-known graph, the synaptic learning rules are always the same. The animal chooses its policy by setting the mode switch that selects one of the available goal signals for the taxis module (*Figure 1*). Nothing has to change under the hood in the operation of the circuit. All the same signals are used for map learning, target learning, and navigation.

In summary, various components of the endotaxis model have appeared in other proposed schemes for spatial learning and navigation. The present model stands out in that all the essential functions are covered in a feed-forward and neuromorphically plausible manner, without invoking unexplained control schemes.

## Animal behavior

The millions of animal species no doubt use a wide range of mechanisms to get around their environment, and it is worth specifying which types of navigation endotaxis might solve. First, the learning mechanism proposed here applies to complex environments, namely those in which discrete paths form sparse connections between points. For a rodent and many other terrestrial animals, the paths they may follow are usually constrained by obstacles or by the need to remain under cover. In those conditions, the brain cannot assume that the distance between points is given by Euclidean geometry, or that beacons for a goal will be visible in a straight line from far away, or that a target can be reached

by following a known heading. As a concrete example, a mouse wishing to exit from deep inside a labyrinth (*Figure 8A*, *Rosenberg et al., 2021*) can draw little benefit from knowing the distance and heading of the entrance.

Second, we are focusing on the early experience with a new environment. Endotaxis can get an animal from zero knowledge to a cognitive map that allows reliable navigation toward goals discovered on a previous foray. It explains how an animal can return home from inside a complex environment on the first attempt (*Rosenberg et al., 2021*) or navigate to a special location after encountering it just once (*Figures 6 and 8*). But it does not implement more advanced routines of spatial reasoning, such as stringing a habitual sequence of actions together into one, or deliberating internally to plan entire routes. Clearly, given enough time in an environment, animals may develop algorithms other than the beginner's choice proposed here.

A key characteristic of endotaxis, distinct from other forms of navigation, is the reliance on trial-and-error. The agent does not deliberate to plan the shortest path to the goal. Instead, it finds the shortest path by locally sampling the real-world actions available at its current point, and choosing the one that maximizes the virtual odor signal. In fact, there is strong evidence that animals navigate by real-world trial-and-error, at least in the early phase of learning (*Redish, 2016*). *Lashley, 1912*, in his first scientific paper on visual discrimination in the rat, reported that rats at a decision point often hesitate 'with a swaying back and forth between the passages.' These actions – called 'vicarious trial and error' – look eerily like sniffing out an odor gradient, but they occur even in the absence of any olfactory cues. Similar behaviors occur in arthropods (*Tarsitano, 2006*) and humans (*Santos-Pata and Verschure, 2018*) when poised at a decision point. We suggest that the animal does indeed sample a gradient, not of an odor, but of an internally generated virtual odor that reflects the proximity to the goal. The animal seems to use the same policy of spatial sampling that it would apply to a real odor signal.

Frequently, a rodent stopped at a maze junction merely turns its head side-to-side, rather than walking down a corridor to sample the gradient. Within the endotaxis model, this could be explained if some of the point cells in the lowest layer (*Figure 1B*) are selective for head direction or for the view down a specific corridor. During navigation, activation of that 'direction cell' systematically precedes activation of point cells further down that corridor. Therefore, the direction cell gets integrated into the map network. From then on, when the animal turns in that direction, this action takes a step along the graph of the environment without requiring a walk in ultimately fruitless directions. In this way, the agent can sample the goal gradient while minimizing energy expenditure.

Once the animal gains familiarity with the environment, it performs fewer of the vicarious trial-and-error movements, and instead moves smoothly through multiple intersections in a row (*Redish, 2016*). This may reflect a transition between different modes of navigation, from the early endotaxis, where every action gets evaluated on its real-world merit, to a mode where many actions are strung together into behavioral motifs. Eventually the animal may also develop an internal forward model for the effects of its own actions, which would allow for prospective planning of an entire route (*Kay et al., 2020*; *Nyberg et al., 2022*). An interesting direction for future research is to seek a neuromorphic circuit model for such action planning; perhaps it can be built naturally on top of the endotaxis circuit.

## Brain circuits

The key elements in the proposed circuitry (*Figure 1*) are a large population of neurons with sparsely selective responses; massive convergence from that population onto a smaller set of output neurons; and synaptic plasticity at the output neurons gated by signals from the animal's experience. A prominent instance of this motif is found in the mushroom body of the arthropod brain (*Heisenberg, 2003*; *Strausfeld et al., 2009*). Here the Kenyon cells, with their sparse odor responses (*Stopfer, 2014*), play the role of both point and map cells. They are strongly recurrently connected; in fact, most of the Kenyon cell output synapses are onto other Kenyon cells (*Eichler et al., 2017*; *Takemura et al., 2017*). Kenyon cells converge onto a much smaller set of mushroom body output neurons (*Aso et al., 2014*), which play the role of goal cells. Plasticity at the synapse between Kenyon cells and output neurons is gated by neuromodulators that encode rewards or punishments (*Cohn et al., 2015*). Mushroom body output neurons are known to guide the turning decisions of the insect (*Aso et al., 2014*), perhaps through their projections to the central complex (*Li et al., 2020*), an area critical to the animal's turning behavior (*Honkanen et al., 2019*). Conceivably, this is where the insect's basic chemotaxis module is implemented.

In the conventional view, the mushroom body helps with odor discrimination and forms memories of discrete odors that are associated with salient experience (*Heisenberg, 2003*). Subsequently, the animal can seek or avoid those odors. But the endotaxis model suggests a different interpretation: insects can also use odors as landmarks in the environment. In this more general form of navigation, the odor is not a goal in itself, but serves to mark a route toward some entirely different goal (*Knaden and Graham, 2016*; *Steck et al., 2009*). A Kenyon cell, through its sparse odor selectivity, may be active at only one place in the environment, and thus provide the required location-selective input to the endotaxis circuit. Recurrent synapses among Kenyon cells will learn the connectivity among these odor-defined locations, and the output neurons will learn to produce a goal signal that leads the insect to a rewarding location, which itself may not even have a defined odor.

Bees and certain ants rely strongly on vision for their navigation. Here the insect uses discrete panoramic views of the landscape as markers for its location (*Webb and Wystrach, 2016*; *Buehlmann et al., 2020*; *Sun et al., 2020*). In those species, the mushroom body receives massive input from visual areas of the brain. If the Kenyon cells respond sparsely to the landscape views, like the point cells in *Figure 1*, then the mushroom body can tie together these discrete vistas into a cognitive map that supports navigation toward arbitrary goal locations.

The same circuit motifs are commonly found in other brain areas, including the mammalian neocortex and hippocampus. While the synaptic circuitry there is less understood than in the insect brain, one can record from neurons more conveniently. Much of that work on neuronal signals during navigation has focused on the rodent hippocampal formation (*Moser et al., 2015*), and it is instructive to compare these recordings to the expectations from the endotaxis model. The three cell types in the model – point cells, map cells, and goal cells – all have place fields, in that they fire preferentially in certain regions within the graph of the environment. However, they differ in important respects.

The place field is smallest for a point cell; somewhat larger for a map cell, owing to recurrent connections in the map network; and larger still for goal cells, owing to additional pooling in the goal network. Such a wide range of place field sizes has indeed been observed in surveys of the rodent hippocampus, spanning at least a factor of 10 in diameter (*Wilson and McNaughton, 1993*; *Kjelstrup et al., 2008*). Some place cells show a graded firing profile that fills the available environment. Furthermore, one finds more place fields near the goal location of a navigation task, even when that location has no overt markers (*Hollup et al., 2001*). Both of those characteristics are expected of the goal cells in the endotaxis model.

The endotaxis model assumes that point cells exist from the very outset in any environment. Indeed, many place cells in the rodent hippocampus appear within minutes of the animal's entry into an arena (*Wilson and McNaughton, 1993*; *Frank et al., 2004*). Furthermore, any given environment activates only a small fraction of these neurons. Most of the 'potential place cells' remain silent, presumably because their sensory trigger feature does not match any of the locations in the current environment (*Alme et al., 2014*; *Epsztein et al., 2011*). In the endotaxis model, each of these sets of point cells is tied into a different map network, which would allow the circuit to maintain multiple cognitive maps in memory (*Muller et al., 1991*).

Goal cells, on the other hand, are expected to have large place fields, centered on a goal location, but extending over much of the environment, so the animal can follow the gradient of their activity (*Burgess and O'Keefe, 1996*). Indeed, such cells have been reported in rat cortex (*Hok et al., 2005*). In the endotaxis model, a goal cell appears suddenly when the animal first arrives at a memorable location, the input synapses from the map network are potentiated, and the neuron immediately develops a place field (*Figure 4*). This prediction is reminiscent of a startling experimental observation in recordings from hippocampal area CA1: a neuron can suddenly start firing with a fully formed place field that may be located anywhere in the environment (*Bittner et al., 2017*). This event appears to be triggered by a calcium plateau potential in the dendrites of the place cell, which potentiates the excitatory synaptic inputs the cell receives. A surprising aspect of this discovery was the large extent of the resulting place field, which requires the animal several seconds to cover. Subsequent cellular measurements indeed revealed a plasticity mechanism that extends over several seconds (*Magee and Grienberger, 2020*). The endotaxis model relies on just such a plasticity rule for map learning (Algorithm 2) that can correlate events at subsequent nodes on the agent's trajectory.

## Outlook

Endotaxis is a hypothetical neural circuit solution to the problems of spatial exploration, learning, and navigation. Its compact circuit structure and all-in-one functionality suggest that it would fit in even the smallest brains. Effectively, endotaxis represents a brain module that could be interposed between a spatial-sensing module, which produces place cells, and a taxis module, which delivers the movements to ascend a goal signal. It further relies on some high-level policy that sets the 'mode switch' by which the animal chooses what goal to pursue. Future research might get at this behavioral control mechanism through a program of anatomical module tracing: first find the neural circuit that controls chemotaxis behavior. Then test if that module receives a convergence of goal signals from other circuits with non-olfactory information. If so, the mechanism of arbitrage that routes one or another goal signal to the taxis module should reveal the high-level coordination of the animal's behavior. Given the recent technical developments in mapping the connectome (*Dorkenwald et al., 2023*), we believe that such a program of module tracing is within reach, probably first for the insect brain.

# Materials and methods

## Simulations

Numerical simulations were performed as described (see Algorithms 1–4). Parameter settings are listed in the text and figure captions. The sensitivity to parameters is reported in *Figure 6*. Code that produced all the results is available in a public repository.

## Average navigated distance

In the text, we often assess the performance of an endotaxis agent by considering point-to-point navigation between all pairs of points on a graph. Given the readout noise $\epsilon$ that affects the goal signal, navigation is a stochastic process with many random decisions along the route. Different random instantiations of the process will produce routes of different lengths. Fortunately, there is a way to calculate the expectation value of the route length without any Monte Carlo simulation.

Consider navigation to goal node $y$. From the state of the network ($\mathbf{M}$ and $\mathbf{G}$), we compute the goal signal $E_{yj}$ at every node $j$. When the agent is at node $j$, it chooses among the neighbor nodes the one with the highest sum of goal signal and noise (Algorithm 1). Based on the goal signal $E_{yj}$ and the noise $\epsilon$, one can compute the probability for each such possible step from $j$. This leads to a transition matrix for the random walk

$$T_{ij}^{(y)} = \text{probability of stepping to } i \text{ when at } j \text{ while in pursuit of } y$$

Subsequent decisions along the route are independent of each other. Hence, the process is a Markov chain. Then we make use of a well-known result for first-capture times on a Markov chain to compute the expected number of steps to arrival at $y$ starting from any node $x$.

Note the method assumes that the process is stationary Markov, such that the goal signal $E_{xy}$ does not change in the course of navigation. In our analysis of patrolling (*Figures 9* and 11), this assumption is violated because the habituation state of the point cells depends on what path the agent took to the current node. In those cases, we resorted to Monte Carlo simulations to estimate the distribution of route lengths.

## Nonlinear activation function

The activation function of a map neuron is the relationship of input to output

$$v_i = f(w_i) \tag{20}$$

where (*Equation 4*)

$$w_i = u_i + \sum_j M_{ij} v_j \tag{21}$$

is the input to the map neuron. Most of the report assumes a linear activation function (*Equation 3*)

$$f(w) = \gamma w \tag{22}$$

For *Figure 7*, we used a saturating function instead (*Equation 20*):

$$f(w) = \begin{cases} \gamma w, & \text{if } w \leq 1 \\ \gamma, & \text{if } w > 1 \end{cases} \tag{23}$$

The recurrent network equation $v_i = f\left(u_i + \sum_j M_{ij} v_j\right)$ was solved using Python's `fsolve`.

## Forgetting of links and resources

In section 'Acquisition of map and targets during exploration,' we discuss the learning algorithm that acquires the connectivity of the environment and the locations of resources. It reacts rapidly to the appearance of new links in the environment: as soon as the agent travels from one point to another, the synapse between the corresponding map cells gets established. Suppose now that a previously existing link becomes blocked: How can one remove the corresponding synapse from the map? A simple solution would be to let all synapses decay over time, balanced by strengthening whenever a link gets traveled. In that case, the entire map would be forgotten when the animal goes to sleep for a few hours, whereas it is clear that animals retain such maps over many days. Instead, one wants a mode of *active* forgetting: memory of the link from node $i$ to $j$ should be weakened only if the agent find itself at node $i$ and repeatedly chooses not to go to $j$. We formalize this in Algorithm 4, which differs only slightly from Algorithm 2.

---

**Algorithm 4** Learning and forgetting.

---

Parameters: gain $\gamma$, threshold $\theta$, goal-learning rate $\alpha$, forgetting rate $\delta$
Input: adjacency matrix $\mathbf{A}$, resource signals $\mathbf{F}$

$\quad \mathbf{M} \leftarrow 0$        ▷ initiate map synapses at 0
$\quad \mathbf{G} \leftarrow 0$        ▷ initiate goal synapses at 0
$\quad t \leftarrow 0$        ▷ $t$ counts the steps
$\quad s(t) \leftarrow x$        ▷ start random walk at $x$
$\quad$ **while** learning **do**
$\quad\quad t \leftarrow t + 1$
$\quad\quad s(t) \leftarrow$ a random neighbor of $s(t-1)$    ▷ continue the random walk
$\quad\quad u_i(t) \leftarrow \delta_{i,s(t)}$ for every point cell $i$    ▷ point cell output
$\quad\quad \mathbf{v}(t) \leftarrow \left(\frac{1}{\gamma}\mathbf{1} - \mathbf{M}\right)^{-1} \mathbf{u}(t)$    ▷ map cell output
$\quad\quad$ **for** all map cell pairs $(i, j)$ **do**
$\quad\quad\quad$ **if** $v_j(t-1) > \theta$ **then**    ▷ if pre-synaptic high
$\quad\quad\quad\quad$ **if** $v_i(t) > \theta$ **then**    ▷ if post-synaptic also high
$\quad\quad\quad\quad\quad M_{ij}, M_{ji} \leftarrow 1$    ▷ potentiate the synapses
$\quad\quad\quad\quad$ **else**    ▷ if post-synaptic low
$\quad\quad\quad\quad\quad M_{ij} \leftarrow e^{-\delta} M_{ij}$    ▷ depress the synapses
$\quad\quad\quad\quad\quad M_{ji} \leftarrow e^{-\delta} M_{ji}$
$\quad\quad\quad\quad$ **end if**
$\quad\quad\quad$ **end if**
$\quad\quad$ **end for**
$\quad\quad \mathbf{r} \leftarrow \mathbf{G}\mathbf{v}(t)$    ▷ goal signals
$\quad\quad$ **for** every goal neuron $k$ **do**
$\quad\quad\quad D \leftarrow F_{k,s(t)} - r_k$    ▷ difference between resource signal and prediction from the map
$\quad\quad\quad$ **if** $D > 0$ **then**    ▷ if the resource signal exceeds the prediction from the map
$\quad\quad\quad\quad$ **for** every map neuron $j$ **do**
$\quad\quad\quad\quad\quad G_{kj} \leftarrow G_{kj} + \alpha D v_j(t)$    ▷ potentiate goal synapses
$\quad\quad\quad\quad$ **end for**
$\quad\quad\quad$ **else**    ▷ if resource signal less than prediction
$\quad\quad\quad\quad$ **for** every map neuron $j$ **do**
$\quad\quad\quad\quad\quad G_{kj} \leftarrow e^{-\delta v_j} G_{kj}$    ▷ depress goal synapses
$\quad\quad\quad\quad$ **end for**
$\quad\quad\quad$ **end if**
$\quad\quad$ **end for**
$\quad$ **end while**

---

Here the added parameter $\delta$ determines how much a map synapse gets depressed each time the corresponding link is not chosen. Similarly, goal synapses decay if their prediction for a resource exceeds the resource signal received by the goal cell. The synaptic learning rule resembles the BCM

rule (*Bienenstock et al., 1982*): synaptic modification is conditional on presynaptic activity and leads to either potentiation or depression depending on the level of postsynaptic activity.

*Figure 10* illustrates this process with a simulation analogous to *Figure 4*. The agent explores a ring graph by a random walk. At some point, a new link appears clear across the ring. Later on that link

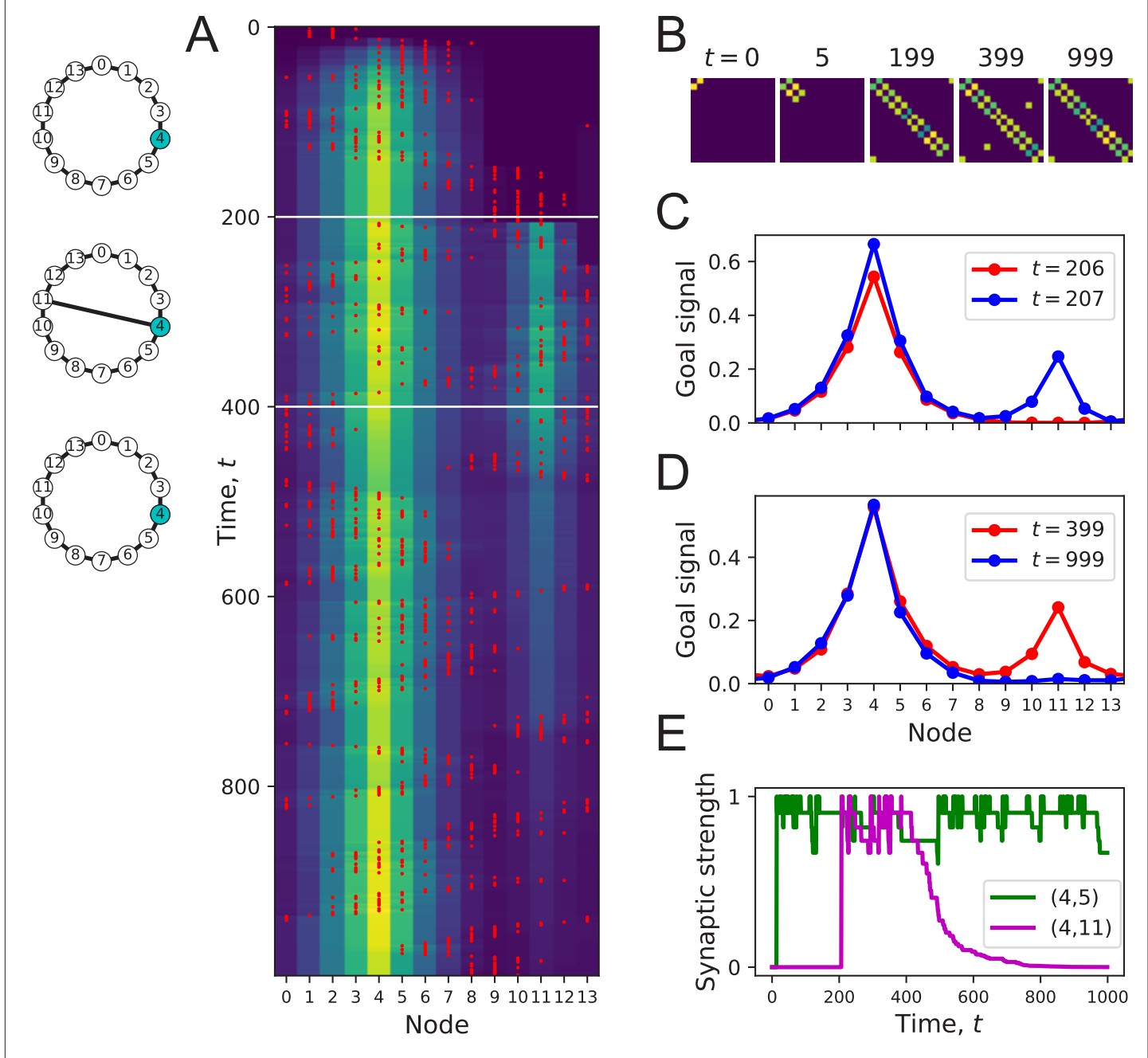

**Figure 10.** Forgetting a link during exploration. (**A**) Simulation of a random walk on a ring with 14 nodes as in *Figure 4*. Left: layout of the ring, with resource locations marked in blue. The walk progresses in 1000 time steps (top to bottom); with the agent's position marked in red (nodes 0–13, horizontal axis). At each time, the color map shows the goal signal that would be produced if the agent were at position 'Node.' White horizontal lines mark the appearance of a new link between nodes 4 and 11 at $t = 200$, and disappearance of that link at $t = 400$. (**B**) The matrix **M** of map synapses at various times. The pixel in row $i$ and column $j$ represents the matrix element $M_{ij}$. Color purple = 0. Note the first few steps (number above graph) each add a new synapse. Eventually, **M** reflects the adjacency matrix of nodes on the graph, and changes as a link is added and removed. (**C**) Goal signals just before and just after the agent travels the new link. (**D**) Goal signals just before the link disappears and at the end of the walk. (**E**) Strength of two synapses in the map, $M_{4,5}$ and $M_{4,11}$, plotted against time during the random walk. Model parameters: $\gamma = 0.32, \theta = 0.27, \alpha = 0.3, \delta = 0.1$.

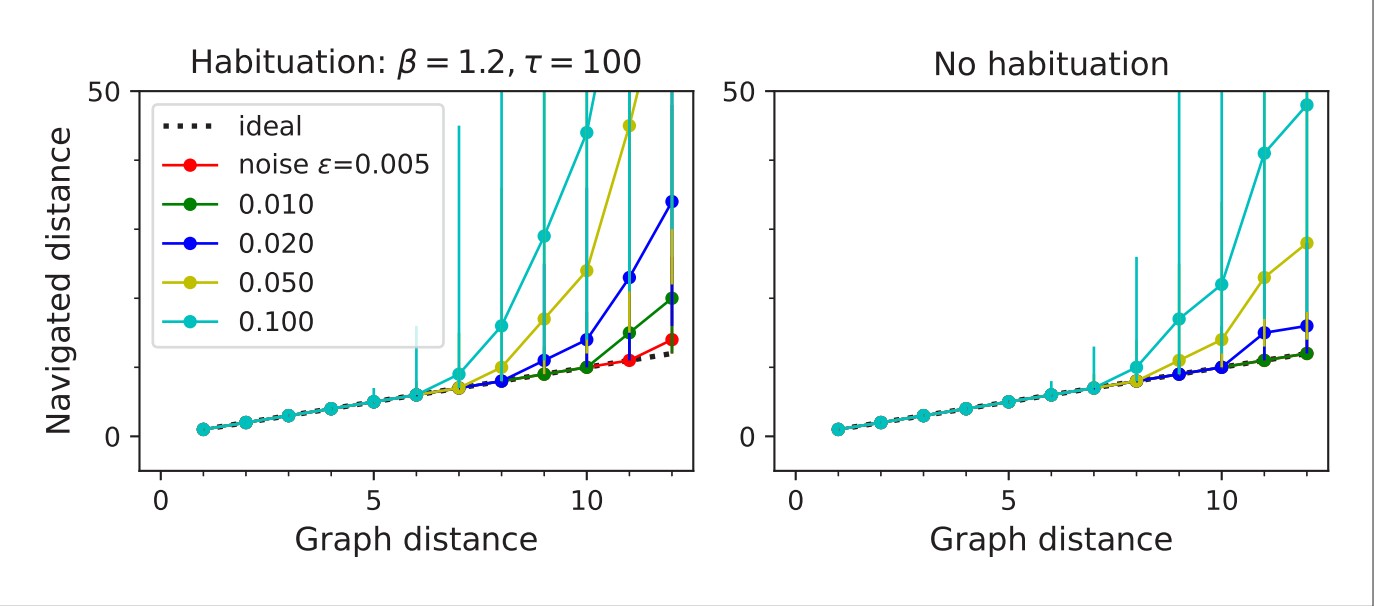

**Figure 11.** Navigation performance with and without habituation. Navigated distance on the binary tree maze, displayed as in *Figure 5E*. Left: an agent with strong habituation: $\beta = 1.2, \tau = 100$. Right: no habituation: $\beta = 0$. The agent learned the map and the goal signals for every node during a random walk with 30,000 steps. Then the agent navigated between all pairs of points on the maze. Graphs show the median ± 10/90 percentile of the navigated distance for all routes with the same graph distance. Other model parameters: $\gamma = 0.33, \theta = 0.30, \alpha = 0.1, \epsilon$ as listed.

disappears again. Acquisition of the link happens very quickly, within a single time step (*Figure 10A and C*). Forgetting that link takes longer, on the order of several hundred steps (*Figure 10A, D and E*). In this simulation, $\delta = 0.1$, so the map synapses decay by about 10% whenever a link is not traveled. One could, of course, accelerate that with a higher $\delta$, but at the cost of destabilizing the entire map. Even the synapses for intact links get depressed frequently (*Figure 10E*) because the random choices of the agent lead it to take any given link only a fraction of the time.

One limitation of the endotaxis agent is that it does not keep a record of what actions are available at each node. Instead, it leaves that information in the environment (see 'Discussion') and simply tries all the actions that are available. When faced with a blocked tunnel, the endotaxis agent does not know that this was previously available. Clearly, a more advanced model of the world that includes a state–action table would allow more effective editing of the cognitive map.

## Habituation in point cells

In section 'Efficient patrolling,' we discuss an extension of the core endotaxis model in which a point neuron undergoes habituation after the agent passes through its node. With every visit, the neuron's sensitivity declines by a factor $e^{-\beta}$. Between visits the sensitivity gradually returns toward 1 with an exponential recovery time of $\tau$ steps (see Algorithm 3).

This addition to the model changes the dynamics of the network input throughout the phases of exploration, navigation, and patrolling. We explored how the resulting performance is affected by applying a strong habituation that decays slowly ($\beta = 1.2, \tau = 100$) and comparing to the basic model with no habituation ($\beta = 0$). During the learning phase, when the map and goal synapses are established via a random walk, the main change is that it takes somewhat longer to learn the map. This is because synaptic updates happen only when both pre- and postsynaptic map cells exceed a threshold (see Algorithm 2), and that requires that both of the respective point neurons be in a high-sensitivity state. Remarkably all the parameter settings ($\gamma, \theta, \alpha$) that support learning and navigating under standard conditions (*Figure 6*) also work well when habituation takes place.

To illustrate the overall effect that habituation has on performance, we simulated learning and navigation on the binary tree graph of *Figure 9*. For every pair of start and end nodes, we asked how the actual navigated distance compared to the shortest graph distance. *Figure 11* shows that performance is affected only slightly. At the standard noise value $\epsilon = 0.01$ used in other simulations, the range of navigation extends over 10 or more steps under both conditions.

## Acknowledgements

This work was supported by the Simons Collaboration on the Global Brain (grant 543015 to MM and 543025 to PP), NSF award 1564330 to PP, and a gift from Google to PP.

## Additional information

### Competing interests

Markus Meister: Reviewing editor, *eLife*. The other authors declare that no competing interests exist.

### Funding

| Funder | Grant reference number | Author |
| --- | --- | --- |
| Simons Foundation | 543015 | Markus Meister |
| Simons Foundation | 543025 | Pietro Perona |
| National Science Foundation | 1564330 | Pietro Perona |
| Google | | Pietro Perona |

The funders had no role in study design, data collection and interpretation, or the decision to submit the work for publication.

### Author contributions

Tony Zhang, Conceptualization, Software, Formal analysis, Investigation, Writing – review and editing; Matthew Rosenberg, Conceptualization, Investigation, Writing – review and editing; Zeyu Jing, Formal analysis, Investigation, Writing – review and editing; Pietro Perona, Conceptualization, Formal analysis, Supervision, Funding acquisition, Investigation, Writing – review and editing; Markus Meister, Conceptualization, Data curation, Software, Formal analysis, Supervision, Funding acquisition, Investigation, Writing - original draft, Writing – review and editing

### Author ORCIDs

Tony Zhang  http://orcid.org/0000-0002-5198-499X
Markus Meister  http://orcid.org/0000-0003-2136-6506

Reviewer #1 (Public Review): https://doi.org/10.7554/eLife.84141.3.sa1
Reviewer #2 (Public Review): https://doi.org/10.7554/eLife.84141.3.sa2
Reviewer #3 (Public Review): https://doi.org/10.7554/eLife.84141.3.sa3
Author Response https://doi.org/10.7554/eLife.84141.3.sa4

## Additional files

### Supplementary files

• MDAR checklist

### Data availability

Data and code to reproduce the reported results are openly available at https://github.com/markus-meister/Endotaxis-2023 (copy archived at *Meister, 2024*).

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
