## [Editor Report · eLife assessment]

This **valuable** work proposes a framework inspired by chemotaxis for understanding how the brain might implement behaviors related to navigating toward a goal. The evidence supporting the conceptual claim is **convincing**. The article proposes a hypothesis that would be of interest to the broad systems neuroscience community, although it was noted the relationship to existing similar hypotheses could be clarified.

---

## [Referee Report · Reviewer #1 (Public Review)]

This paper presents a highly compelling and novel hypothesis for how the brain could generate signals to guide navigation toward remembered goals. Under this hypothesis, which the authors call "Endotaxis", the brain co-opts its ancient ability to navigate up odor gradients (chemotaxis) by generating a "virtual odor" that grows stronger the closer the animal is to a goal location. This idea is compelling from an evolutionary perspective and a mechanistic perspective. The paper is well-written and delightful to read.

The authors develop a detailed model of how the brain may perform "Endotaxis", using a variety of interconnected cell types (point, map, and goal cells) to inform the chemotaxis system. They tested the ability of this model to navigate in several state spaces, representing both physical mazes and abstract cognitive tasks. The Endotaxis model performed reasonably well across different environments and different types of goals.

The authors further tested the model using parameter sweeps and discovered a critical level of network gain, beyond which task performance drops. This critical level approximately matched analytical derivations.

Overall, this paper provides a very compelling model for how neural circuits may have evolved the ability to navigate towards remembered goals, using ancient chemotaxis circuits.

This framework will likely be very important for understanding how the hippocampus (and other memory/navigation-related circuits) interfaces with other processes in the brain, giving rise to memory-guided behavior.

---

## [Referee Report · Reviewer #2 (Public Review)]

The manuscript presents a computational model of how an organism might learn a map of the structure of its environment and the location of valuable resources through synaptic plasticity, and how this map could subsequently be used for goal-directed navigation.

The model is composed of 'map cells', which learn the structure of the environment in their recurrent connections, and 'goal-cell' which store the location of valued resources with respect to the map cell population. Each map cell corresponds to a particular location in the environment due to receiving external excitatory input at this location. The synaptic plasticity rule between map cells potentiates synapses when activity above a specified threshold at the pre-synaptic neuron is followed by above-threshold activity at the post-synaptic neuron. The threshold is set such that map neurons are only driven above this plasticity threshold by the external excitatory input, causing synapses to only be potentiated between a pair of map neurons when the organism moves directly between the locations they represent. This causes the weight matrix between the map neurons to learn the adjacency for the graph of locations in the environment, i.e. after learning the synaptic weight matrix matches the environment's adjacency matrix. Recurrent activity in the map neuron population then causes a bump of activity centred on the current location, which drops off exponentially with the diffusion distance on the graph. Each goal cell receives input from the map cells, and also from a 'resource cell' whose activity indicates the presence or absence of a given values resource at the current location. Synaptic plasticity potentiates map-cell to goal-cell synapses in proportion to the activity of the map cells at time-points when the resource cell is active. This causes goal cell activity to increase when the activity of the map cell population is similar to the activity where the resource was obtained. The upshot of all this is that after learning the activity of goal cells decreases exponentially with the diffusion distance from the corresponding goal location. The organism can therefore navigate to a given goal by doing gradient ascent on the activity of the corresponding goal cell. The process of evaluating these gradients and using them to select actions is not modelled explicitly, but the authors point to the similarity of this mechanism to chemotaxis (ascending a gradient of odour concentration to reach the odour source), and the widespread capacity for chemotaxis in the animal kingdom, to argue for its biological plausibility. The ideas are interesting and the presentation of the results in the manuscript is generally clear.

Closely related ideas have been explored in previous work, and there are some aspects of how the work relates to previous literature that it would be useful to clarify. Several lines of work have proposed learning long-range relationships between states in the environment, to enable navigation to rewarding goals by effectively descending distance gradients. The most well-known of these in the neuroscience literature is the Successor Representation (SR) (Dayan 1993), which is defined as the expected discounted future occupancy of each state given the current state. As noted in the discussion, this is closely related to the representation learnt by the map cells in the current model. The key difference is that the successor representation uses state-state transitions under a given policy (a mapping from states to actions), whereas the current model uses the adjacency matrix between states, which depends only on the environment and hence is independent of the policy followed while the representation is learnt (given sufficient exploration). This policy independence is useful, as the SR can fail to generate good routes to goals when these are very different from the policy under which it was learned (see Russek et al. https://doi.org/10.1371/journal.pcbi.1005768). However, there are several prior proposals for policy-independent SR-like mechanisms that it would be useful to discuss. Baram et al. (https://doi.org/10.1101/421461) propose navigating to goals by doing gradient descent on diffusion distances, computed as powers of the adjacency matrix as in the current work. One limitation of using the adjacency matrix is that it does not handle situations where transitions between states are probabilistic, which is not a big issue for navigation in physical space but is for applying the mechanism to cognitive tasks more broadly. There are prior ideas for learning policy-independent representations similar to the SR that do not have this limitation. Kaelbling (Learning to achieve goals, IJCAI, 1993) proposed using an off-policy learning rule similar to Q-learning, to learn shortest path distances between states. Piray and Daw (https://doi.org/10.1038/s41467-021-25123-3) consider a default representation, which is a successor-like representation under a generic default policy, building on the Linear Markov Decision Process (LMDP) framework of Todorov (https://doi.org/10.1073/pnas.0710743106). Also relevant to the current study is the work of Fang et al. (https://doi.org/10.7554/eLife.80680) who, as in the current work, propose using recurrent network dynamics to compute a long-range representation (the SR) from synaptic weights that store local transition information.

One other area where I felt the work could be better integrated with the existing literature was the discussion of mapping the model onto brain circuits. An interesting and attractive aspect of the work is the idea that the relatively high-level operation of goal-directed navigation could be built on top of evolutionarily older mechanisms for ascending odour gradients. Given this framing, I was expecting the discussion of brain circuits to consider interactions between spatial mapping systems and regions involved in olfactory processing. However the discussion of mammalian brains focussed exclusively on the hippocampus without any link to olfaction, which feels like a missed opportunity. I am not an expert on olfaction, but one region that seems particularly interesting in this context is the olfactory tubercle (see Wesson & Wilson https://doi.org/10.1016/j.neubiorev.2010.08.004 for a review). This region is contiguous with the ventral striatum and has similar local circuitry, receives strong input from olfactory regions, but also input from the hippocampal formation, and a strong dopaminergic innervation from VTA. This suggests a mapping of the model to brain circuits in which map cells in the hippocampal formation project to goal cells in the olfactory tubercle, with the dopaminergic input acting as resource cells (note that different dopamine neuron populations appear to respond to different reward types, see e.g. https://doi.org/10.1038/s41586-022-04954-0, https://doi.org/10.1101/2023.05.09.540067). I was also surprised not to see any discussion of internally generated sequential activity in the hippocampus as a possible mechanism for the look-ahead needed to evaluate the goal distance gradient, particularly given the authors suggest that vicarious trial and error (VTE) is a behavioural signature of this gradient sampling, and it is known that during VTE hippocampus plays out internally generated sequences of possible future locations (see Redish https://doi.org/10.1038/nrn.2015.30).

---

## [Referee Report · Reviewer #3 (Public Review)]

This paper describes an algorithm that provides a general mechanism for goal-directed behaviour in a biologically plausible neural form.

The method depends on substantial simplifying assumptions. The simulated animal effectively moves through an environment consisting of discrete locations and can reliably detect when it is in each location. Whenever it moves from one location to an adjacent location, it perfectly learns the connectivity between these two locations (changes the value in an adjacency matrix to 1). This creates a graph of connections that reflects the explored environment. In this graph, the current location gets input activation and this spreads to all connected nodes multiplied by a constant decay (adjusted to the branching number of the graph) so that as the number of connection steps increases the activation decreases. Some locations will be marked as goals through experiencing a resource of a specific identity there and subsequently will be activated by an amount proportional to their distance in the graph from the current location, i.e., their activation will increase if the agent moves a step closer and decrease if it moves a step further away. Hence by making such exploratory movements, the animal can decide which way to move to obtain a specified goal.

Although the algorithm is presented within a conceptual framework of chemotaxis, I.e., making movements to sample a local gradient and move up it, the approach relates closely to previous models of exploration, learning, and navigation that similarly establish (through experience) a graph structure to represent how locations are connected and use some form of activity-propagation from the current node or goal node to identify a (shortest) route between them. Many of these similarly claim to be plausible neural circuits. The current authors argue that the current algorithm has several desirable features with respect to such previous work: for example, the 'readout' of the path does not require explicit 'look-up' and activation of the goal node (although it does require a choice of which goal node is currently connected to behavior); and does not require any separate control or rules for learning vs. navigation phases. By comparison to the successor representation method used in RL, which also appears related, they note that the gain (decay) factor is not equivalent to a temporal discount and that their method learns only state-state transitions, allowing the value of actions to be externalised, I.e., calculated by trying alternative actions to see which increases the activation at the goal node the most. On the other hand, it should be noted that some issues addressed in previous models, such as uncertainty over the current state or probabilistic state(-action) transitions are not addressed in this work.

The algorithm presents some elegant features with respect to previous work such as conceptually separating the 'goal' nodes from the state (location) graph (I.e. 'goals' are not just special target states within the graph) so that a small number of goals can become associated to (potentially) multiple regions of the state graph where they are satisfied, or near to being satisfied. This architecture is suggested, in the discussion, to resemble the insect mushroom body (MB), where it is known that a small number of output neurons (MBONs, putative goal neurons) are activated by plastic connections from Kenyon cells (KCs, putative state neurons). However, it goes substantially beyond any available evidence to claim that KC connectivity could support the acquisition of a graph (in the form of an adjacency matrix) representing the structure of the environment: KCs show sparse distributed activity (not one active node per state); it seems unlikely that any two arbitrary KCs can (rapidly) become connected; and as yet has not been demonstrated that KC connectivity is plastic at all.

The results presented are fairly straightforward given the simplification of the tasks, as described above. They show (1) in practical terms, the spreading signal travels further for a larger decay but becomes erratic as the decay parameter (map neuron gain) approaches its theoretical upper bound and decreases below noise levels beyond a certain distance. Both follow the theory but it is perhaps helpful to see that there is a viable range of values of the gain for which the mechanism works, that is, it is not highly dependent on precise tuning. (2) That different graph structures can be acquired and used to approach goal locations (not surprising). (3) That simultaneous learning and exploitation of the graph only minimally affects the performance over starting with perfect knowledge of the graph. (4) That the parameters interact in expected ways. (5) That the separation of goals from states can be used flexibly e.g. the homing behaviour (a goal state is learned before any of the map is learned) and the patrolling behaviour (a goal cell that monitors all states for how recently they were visited). It is also interesting to link the mechanism of exploration of neighbouring states to observed scanning behaviours in navigating animals. It would have been interesting to explore whether the parameters could be dynamically tuned, based on the overall graph activity.

---

## [Author Response]

The following is the authors’ response to the original reviews.

Thank you for the detailed and constructive reviews. We revised the paper accordingly, and a point-by-point reply appears below. The main changes are:

An extended discussion section that places our work in context with other related developments in theory and modeling.A new results section that demonstrates a substantial improvement in performance from a non-linear activation function. This led to addition of a co-author.The mathematical proof that the resolvent of the adjacency matrix leads to the shortest path distances has been moved to a separate article, available as a preprint and attached to this resubmission. This allows us to present that work in the context of graph theory, and focus the present paper on neural modeling.

**Reviewer #1 (Public Review):**
This paper presents a highly compelling and novel hypothesis for how the brain could generate signals to guide navigation towards remembered goals. Under this hypothesis, which the authors call "Endotaxis", the brain co-opts its ancient ability to navigate up odor gradients (chemotaxis) by generating a "virtual odor" that grows stronger the closer the animal is to a goal location. This idea is compelling from an evolutionary perspective and a mechanistic perspective. The paper is well-written and delightful to read.The authors develop a detailed model of how the brain may perform "Endotaxis", using a variety of interconnected cell types (point, map, and goal cells) to inform the chemotaxis system. They tested the ability of this model to navigate in several state spaces, representing both physical mazes and abstract cognitive tasks. The Endotaxis model performed reasonably well across different environments and different types of goals.The authors further tested the model using parameter sweeps and discovered a critical level of network gain, beyond which task performance drops. This critical level approximately matched analytical derivations.My main concern with this paper is that the analysis of the critical gain value (gamma_c) is incomplete, making the implications of these analyses unclear. There are several different reasonable ways in which the Endotaxis map cell representations might be normalized, which I suspect may lead to different results. Specifically, the recurrent connections between map cells may either be an adjacency matrix, or a normalized transition matrix. In the current submission, the recurrent connections are an unnormalized adjacency matrix. In a previous preprint version of the Endotaxis manuscript, the recurrent connections between the map cells were learned using Oja's rule, which results in a normalized state-transition matrix (see "Appendix 5: Endotaxis model and the successor representation" in "Neural learning rules for generating flexible predictions and computing the successor representation", your reference 17). The authors state "In summary, this sensitivity analysis shows that the optimal parameter set for endotaxis does depend on the environment". Is this statement, and the other conclusions of the sensitivity analysis, still true if the learned recurrent connections are a properly normalized state-transition matrix?

Yes, this is an interesting topic. In v.1 of our bioRxiv preprint we used Oja’s rule for learning, which will converge on a map connectivity that reflects the transition probabilities. The matrix M becomes a left-normalized or right-normalized stochastic matrix, depending on whether one uses the pre-synaptic or the post-synaptic version of Oja’s rule. This is explained well in Appendix 5 of Fang 2023.

In the present version of the model we use a rule that learns the adjacency matrix A, not the transition matrix T. The motivation is that we want to explain instances of oneshot learning, where an agent acquires a route after traversing it just once. For example, we had found experimentally that mice can execute a complex homing route on the first attempt.

An agent can establish whether two nodes are connected (adjacency) the very first time it travels from one node to the other. Whereas it can evaluate the transition probability for that link only after trying this and all the other available links on multiple occasions. Hence the normalization terms in Oja’s rule, or in the rule used by Fang 2023, all involve some time-averaging over multiple visits to the same node. This implements a gradual learning process over many experiences, rather than a one-shot acquisition on the first experience.

Still one may ask whether there are advantages to learning the transition matrix rather than the adjacency matrix. We looked into this with the following results:

• The result that (1/γ − A)−1 is monotonically related to the graph distances D in the limit of small γ (a proof now moved to the Meister 2023 preprint) , holds also for thetransition matrix T. The proof follows the same steps. So in the small gain limit, the navigation model would work with T as well.

• If one uses the transition matrix to compute the network output (1/γ − T)-1 then the critical gain value is γc = 1. It is well known that the largest eigenvalue of any Markov transition matrix is 1, and the critical gain γc is the inverse of that. This result is independent of the graph. So this offers the promise that the network could use the same gain parameter γ regardless of the environment.

• In practice, however, the goal signal turned out to be less robust when based on T than when based on A. We illustrate this with the attached Author response image 1. This replicates the analysis in Figure 3 of the manuscript, using the transition matrix instead of the adjacency matrix. Some observations:

• Panel B: The goal signal follows an exponential dependence on graph distance much more robustly for the model with A than with T. This holds even for small gain values where the exponential decay is steep.

• Panel C: As one raises the gain closer to the critical value, the goal signal based on T scatters much more than when based on A.

• Panels D, E: Navigation based on A works better than based on T. For example, using the highest practical gain value, and a readout noise of ϵ = 0.01, navigation based on T has a range of only 8 steps on this graph, whereas navigation based on A ranges over 12 steps, the full size of this graph.

We have added a section “Choice of learning rule” to explain this. The Author response image 1 is part of the code notebook on Github.

**Author response image 1. sa4fig1:** 

Overall, this paper provides a very compelling model for how neural circuits may have evolved the ability to navigate towards remembered goals, using ancient chemotaxis circuits.This framework will likely be very important for understanding how the hippocampus (and other memory/navigation-related circuits) interfaces with other processes in the brain, giving rise to memory-guided behavior.
**Reviewer #2 (Public Review):**
The manuscript presents a computational model of how an organism might learn a map of the structure of its environment and the location of valuable resources through synaptic plasticity, and how this map could subsequently be used for goal-directed navigation.The model is composed of 'map cells', which learn the structure of the environment in their recurrent connections, and 'goal-cell' which stores the location of valued resources with respect to the map cell population. Each map cell corresponds to a particular location in the environment due to receiving external excitatory input at this location. The synaptic plasticity rule between map cells potentiates synapses when activity above a specified threshold at the pre-synaptic neuron is followed by above-threshold activity at the post-synaptic neuron. The threshold is set such that map neurons are only driven above this plasticity threshold by the external excitatory input, causing synapses to only be potentiated between a pair of map neurons when the organism moves directly between the locations they represent. This causes the weight matrix between the map neurons to learn the adjacency for the graph of locations in the environment, i.e. after learning the synaptic weight matrix matches the environment's adjacency matrix. Recurrent activity in the map neuron population then causes a bump of activity centred on the current location, which drops off exponentially with the diffusion distance on the graph. Each goal cell receives input from the map cells, and also from a 'resource cell' whose activity indicates the presence or absence of a given values resource at the current location. Synaptic plasticity potentiates map-cell to goal-cell synapses in proportion to the activity of the map cells at time points when the resource cell is active. This causes goal cell activity to increase when the activity of the map cell population is similar to the activity where the resource was obtained. The upshot of all this is that after learning the activity of goal cells decreases exponentially with the diffusion distance from the corresponding goal location. The organism can therefore navigate to a given goal by doing gradient ascent on the activity of the corresponding goal cell. The process of evaluating these gradients and using them to select actions is not modelled explicitly, but the authors point to the similarity of this mechanism to chemotaxis (ascending a gradient of odour concentration to reach the odour source), and the widespread capacity for chemotaxis in the animal kingdom, to argue for its biological plausibility.The ideas are interesting and the presentation in the manuscript is generally clear. The two principle limitations of the manuscript are: (i) Many of the ideas that the model implements have been explored in previous work. (ii) The mapping of the circuit model onto real biological systems is pretty speculative, particularly with respect to the cerebellum.Regarding the novelty of the work, the idea of flexibly navigating to goals by descending distance gradients dates back to at least Kaelbling (Learning to achieve goals, IJCAI, 1993), and is closely related to both the successor representation (cited in manuscript) and Linear Markov Decision Processes (LMDPs) (Piray and Daw, 2021, https://doi.org/ 10.1038/s41467-021-25123-3, Todorov, 2009 https://doi.org/10.1073/pnas.0710743106). The specific proposal of navigating to goals by doing gradient descent on diffusion distances, computed as powers of the adjacency matrix, is explored in Baram et al. 2018 (https://doi.org/10.1101/421461), and the idea that recurrent neural networks whose weights are the adjacency matrix can compute diffusion distances are explored in Fang et al. 2022 (https://doi.org/10.1101/2022.05.18.492543). Similar ideas about route planning using the spread of recurrent activity are also explored in Corneil and Gerstner (2015, cited in manuscript). Further exploration of this space of ideas is no bad thing, but it is important to be clear where prior literature has proposed closely related ideas.

We have added a discussion section on “Theories and models of spatial learning” with a survey of ideas in this domain and how they come together in the Endotaxis model.

Regarding whether the proposed circuit model might plausibly map onto a real biological system, I will focus on the mammalian brain as I don't know the relevant insect literature. It was not completely clear to me how the authors think their model corresponds to mammalian brain circuits. When they initially discuss brain circuits they point to the cerebellum as a plausible candidate structure (lines 520-546). Though the correspondence between cerebellar and model cell types is not very clearly outlined, my understanding is they propose that cerebellar granule cells are the 'map-cells' and Purkinje cells are the 'goal-cells'. I'm no cerebellum expert, but my understanding is that the granule cells do not have recurrent excitatory connections needed by the map cells. I am also not aware of reports of place-field-like firing in these cell populations that would be predicted by this correspondence. If the authors think the cerebellum is the substrate for the proposed mechanism they should clearly outline the proposed correspondence between cerebellar and model cell types and support the argument with reference to the circuit architecture, firing properties, lesion studies, etc.

On further thought we agree that the cerebellum-like circuits are not a plausible substrate for the endotaxis algorithm. The anatomy looks compelling, but plasticity at the synapse is anti-hebbian, and - as the reviewer points out - there is little evidence for recurrence among the inputs. We changed the discussion text accordingly.

The authors also discuss the possibility that the hippocampal formation might implement the proposed model, though confusingly they state 'we do not presume that endotaxis is localized to that structure' (line 564).

We have removed that confusing bit of text.

A correspondence with the hippocampus appears more plausible than the cerebellum, given the spatial tuning properties of hippocampal cells, and the profound effect of lesions on navigation behaviours. When discussing the possible relationship of the model to hippocampal circuits it would be useful to address internally generated sequential activity in the hippocampus. During active navigation, and when animals exhibit vicarious trial and error at decision points, internally generated sequential activity of hippocampal place cells appears to explore different possible routes ahead of the animal (Kay et al. 2020, https://doi.org/10.1016/j.cell.2020.01.014, Reddish 2016, https:// doi.org/10.1038/nrn.2015.30). Given the emphasis the model places on sampling possible future locations to evaluate goal-distance gradients, this seems highly relevant.

In our model, the possible future locations are sampled in real life, with the agent moving there or at least in that direction, e.g. via VTE movements. In this simple form the model has no provision for internal planning, and the animal never learns any specific route sequence. One can envision extending such a model with some form of sequence learning that would then support an internal planning mechanism. We mention this in the revised discussion section, along with citation of these relevant articles.

Also, given the strong emphasis the authors place on the relationship of their model to chemotaxis/odour-guided navigation, it would be useful to discuss brain circuits involved in chemotaxis, and whether/how these circuits relate to those involved in goal-directed navigation, and the proposed model.

The neural basis of goal-directed navigation is probably best understood in the insect brain. There the locomotor decisions seem to be initiated in the central complex, whose circuitry is getting revealed by the fly connectome projects. This area receives input from diverse sensory areas that deliver the signal on which the decisions are based. That includes the mushroom body, which we argue has the anatomical structure to implement the endotaxis algorithm. It remains a mystery how the insect chooses a particular goal for pursuit via its decisions. It could be revealing to force a change in goals (the mode switch in the endotaxis circuit) while recording from brain areas like the central complex.Our discussion now elaborates on this.

Finally, it would be useful to clarify two aspects of the behaviour of the proposed algorithm:1. When discussing the relationship of the model to the successor representation (lines 620-627), the authors emphasise that learning in the model is independent of the policy followed by the agent during learning, while the successor representation is policy dependent. The policy independence of the model is achieved by making the synapses between map cells binary (0 or 1 weight) and setting them to 1 following a single transition between two locations. This makes the model unsuitable for learning the structure of graphs with probabilistic transitions, e.g. it would not behave adaptively in the widely used two-step task (Daw et al. 2011, https://doi.org/10.1016/j.neuron.2011.02.027) as it would fail to differentiate between common and rare transitions. This limitation should be made clear and is particularly relevant to claims that the model can handle cognitive tasks in general. It is also worth noting that there are algorithms that are closely related to the successor representation, but which learn about the structure of the environment independent of the subjects policy, e.g. the work of Kaelbling which learns shortest path distances, and the default representation in the work of Piray and Daw (both referenced above). Both these approaches handle probabilistic transition structures.

Yes. Our problem statement assumes that the environment is a graph with fixed edge weights. The revised text mentions this and other assumptions in a new section “Choice of learning rule”.

1. As the model evaluates distances using powers of adjacency matrix, the resulting distances are diffusion distances not shortest path distances. Though diffusion and shortest path distances are usually closely correlated, they can differ systematically for some graphs (see Baram et al. ci:ted above).

The recurrent network of map cells implements a specific function of the adjacency matrix, namely the resolvent (Eqn 7). We have a mathematical proof that this function delivers the shortest graph distances exactly, in the limit of small gain (γ in Eqn 7), and that this holds true for all graphs. For practical navigation in the presence of noise, one needs to raise the gain to something finite. Figure 3 analyzes how this affects deviations from the shortest graph distance, and how nonetheless the model still supports effective navigation over a surprising range. The mathematical details of the proof and further exploration of the resolvent distance at finite gain have been moved to a separate article, which is cited from here, and attached to the submission. The preprint by Baram et al. is cited in that article.

**Reviewer #3 (Public Review):**
This paper argues that it has developed an algorithm conceptually related to chemotaxis that provides a general mechanism for goal-directed behaviour in a biologically plausible neural form.The method depends on substantial simplifying assumptions. The simulated animal effectively moves through an environment consisting of discrete locations and can reliably detect when it is in each location. Whenever it moves from one location to an adjacent location, it perfectly learns the connectivity between these two locations (changes the value in an adjacency matrix to 1). This creates a graph of connections that reflects the explored environment. In this graph, the current location gets input activation and this spreads to all connected nodes multiplied by a constant decay (adjusted to the branching number of the graph) so that as the number of connection steps increases the activation decreases. Some locations will be marked as goals through experiencing a resource of a specific identity there, and subsequently will be activated by an amount proportional to their distance in the graph from the current location, i.e., their activation will increase if the agent moves a step closer and decrease if it moves a step further away. Hence by making such exploratory movements, the animal can decide which way to move to obtain a specified goal.I note here that it was not clear what purpose, other than increasing the effective range of activation, is served by having the goal input weights set based on the activation levels when the goal is obtained. As demonstrated in the homing behaviour, it is sufficient to just have a goal connected to a single location for the mechanism to work (i.e., the activation at that location increases if the animal takes a step closer to it); and as demonstrated by adding a new graph connection, goal activation is immediately altered in an appropriate way to exploit a new shortcut, without the goal weights corresponding to this graph change needing to be relearnt.

As the reviewer states, allowing a graded strengthening of multiple synapses from the map cells increases the effective range of the goal signal. We have now confirmed this in simulations. For example, in the analysis of Fig 3E, a single goal synapse enables perfect navigation only over a range of 7 steps, whereas the distributed goal synapses allow perfect navigation over the full 12 steps. This analysis is included in the code notebook on Github.

Given the abstractions introduced, it is clear that the biological task here has been reduced to the general problem of calculating the shortest path in a graph. That is, no real-world complications such as how to reliably recognise the same location when deciding that a new node should be introduced for a new location, or how to reliably execute movements between locations are addressed. Noise is only introduced as a 1% variability in the goal signal. It is therefore surprising that the main text provides almost no discussion of the conceptual relationship of this work to decades of previous work in calculating the shortest path in graphs, including a wide range of neural- and hardwarebased algorithms, many of which have been presented in the context of brain circuits.The connection to this work is briefly made in appendix A.1, where it is argued that the shortest path distance between two nodes in a directed graph can be calculated from equation 15, which depends only on the adjacency matrix and the decay parameter (provided the latter falls below a given value). It is not clear from the presentation whether this is a novel result. No direct reference is given for the derivation so I assume it is novel. But if this is a previously unknown solution to the general problem it deserves to be much more strongly featured and either way it needs to be appropriately set in the context of previous work.

As far as we know this proposal for computing all-pairs-shortest-path is novel. We could not find it in textbooks or an extended literature search. We have discussed it with two graph theorist colleagues, who could not recall seeing it before, although the proof of the relationship is elementary. Inspired by the present reviewer comment, we chose to publish the result in a separate article that can focus on the mathematics and place it in the appropriate context of prior work in graph theory. For related work in the area of neural modeling please see our revised discussion section.

Once this principle is grasped, the added value of the simulated results is somewhat limited. These show: (1) in practical terms, the spreading signal travels further for a smaller decay but becomes erratic as the decay parameter (map neuron gain) approaches its theoretical upper bound and decreases below noise levels beyond a certain distance. Both follow the theory. (2) that different graph structures can be acquired and used to approach goal locations (not surprising) . (3) that simultaneous learning and exploitation of the graph only minimally affects the performance over starting with perfect knowledge of the graph. (4) that the parameters interact in expected ways. It might have been more impactful to explore whether the parameters could be dynamically tuned, based on the overall graph activity.

This is a good summary of our simulation results, but we differ in the assessment of their value. In our experience, simulations can easily demolish an idea that seemed wonderful before exposure to numerical reality. For example, it is well known that one can build a neural integrator from a recurrent network that has feedback gain of exactly 1. In practical simulations, though, these networks tend to be fickle and unstable, and require unrealistically accurate tuning of the feedback gain. In our case, the theory predicts that there is a limited range of gains that should work, below the critical value, but large enough to avoid excessive decay of the signal. Simulation was needed to test what this practical range was, and we were pleasantly surprised that it is not ridiculously small, with robust navigation over a 10-20% range. Similarly, we did not predict that the same parameters would allow for effective acquisition of a new graph, learning of targets within the graph, and shortest-route navigation to those targets, without requiring any change in the operation of the network.

Perhaps the most biologically interesting aspect of the work is to demonstrate the effectiveness, for flexible behaviour, of keeping separate the latent learning of environmental structure and the association of specific environmental states to goals or values. This contrasts (as the authors discuss) with the standard reinforcement learning approach, for example, that tries to learn the value of states that lead to reward. Examples of flexibility include the homing behaviour (a goal state is learned before any of the map is learned) and the patrolling behaviour (a goal cell that monitors all states for how recently they were visited). It is also interesting to link the mechanism of exploration of neighbouring states to observed scanning behaviours in navigating animals.The mapping to brain circuits is less convincing. Specifically, for the analogy to the mushroom body, it is not clear what connectivity (in the MB) is supposed to underlie the graph structure which is crucial to the whole concept. Is it assumed that Kenyon cell connections perform the activation spreading function and that these connections are sufficiently adaptable to rapidly learn the adjacency matrix? Is there any evidence for this?

Yes, there is good evidence for recurrent synapses among Kenyon cells (map cells in the model), and for reward-gated synaptic plasticity at the synapses onto mushroom body output cells (goal cells in our model). We have expanded this material in the discussion section. Whether those functions are sufficient to learn the structure of a spatial environment has not been explored; we hope our paper might give an impetus, and are exploring behavioral experiments on flies with colleagues.

As discussed above, the possibility that an algorithm like 'endotaxis' could explain how the rodent place cell system could support trajectory planning has already been explored in previous work so it is not clear what additional insight is gained from the current model.

Please see our revised discussion section on “theories and models of spatial learning”.In short, some ingredients of the model have appeared in prior work, but we believe that the present formulation offers an unexpectedly simple end-to-end solution for all components of navigation: exploration, target learning, and goal seeking.

**Reviewer #1 (Recommendations For The Authors):**
Major concern:See the public review. How do the results change depending on whether the recurrent connections between map cells are an adjacency matrix vs. a properly normalized statetransition matrix? I'm especially asking about results related to critical gain (gamma_c), and the dependence of the optimal parameter values on the environment.

Please see our response above including the attached reviewer figure.

Minor concerns:It is not always clear when the learning rule is symmetric vs asymmetric (undirected vs directed graph), and it seems to switch back and forth. For example, line 127 refers to a directed graph; Fig 2B and the intro describe symmetric Hebbian learning. Most (all?) of the simulations use the symmetric rule. Please make sure it's clear.

For simplicity we now use a symmetric rule throughout, as is appropriate for undirected graphs. We mention that a directed learning rule could be used to learn directed graphs.See the section on “choice of learning rule”.M_ij is not defined when it's first introduced (eq 4). Consider labeling the M's and the G's in Fig 2.

Done.

The network gain factor (gamma, eq 4) is distributed over both external and recurrent inputs (v = gamma(u + Mv)), instead of local to the recurrent weights like in the Successor Representation. This notational choice is obviously up to the authors. I raise slight concern for two reasons -- first, distributing gamma may affect some of the parameter sweep results (see major concern), and second, it may be confusing in light of how gamma is used in the SR literature (see reviewer's paper for the derivation of how SR is computed by an RNN with gain gamma).

In our model, gamma represents the (linear) activation function of the map neuron, from synaptic input to firing output. Because the synaptic input comes from point cells and also from other map cells, the gain factor is applied to both. See for example the Dayan & Abbott book Eqn 7.11, which at steady state becomes our Eqn 4. In the formalism of Fang 2023 (Eqn 2), the factor γ is only applied to the recurrent synaptic input J ⋅ f, but somehow not to the place cell input ϕ. Biophysically, one could imagine applying the variable gain only to the recurrent synapses and not the feed-forward ones. Instead we prefer to think of it as modulating the gain of the neurons, rather than the synapses. The SR literature follows conventions from the early reinforcement learning papers, which were unconstrained by thinking about neurons and synapses. We have added a footnote pointing the reader to the uses of γ in different papers.

In eq 13, and simulations, noise is added to the output only, not to the activity of recurrently connected neurons. It is possible this underestimates the impact of noise since the same magnitude of noise in the recurrent network (map cells) could have a compounded effect on the output.

Certainly. The equivalent output noise represents the cumulative effect of noise everywhere in the network. We argue that a cumulative effect of 1% is reasonable given the overall ability of animals at stimulus discrimination, which is also limited by noise everywhere in the network. This has been clarified in the text.

Fig 3 E, F, it looks like the navigated distance may be capped. I ask because the error bars for graph distance = 12 are so small/nonexistent. If it's capped, this should be in the legend.

Correct. 12 is the largest distance on this graph. This has been added to the caption.

Fig 3D legend, what does "navigation failed" mean? These results are not shown.

On those occasions the agent gets trapped at a local maximum of the goal signal other than the intended goal. We have removed that line as it is not needed to interpret the data.

Line 446, typo (Lateron).

Fixed.

Line 475, I'm a bit confused by the discussion of birds and bats. Bird behavior in the real world does involve discrete paths between points. Even if they theoretically could fly between any points, there are costs to doing so, and in practice, they often choose discrete favorite paths. It is definitely plausible that animals that can fly could also employ Endotaxis, so it is confusing to suggest they don't have the right behavior for Endotaxis, especially given the focus on fruit flies later in the discussion.

Good points, we removed that remark. Regarding fruit flies, they handle much important business while walking, such as tracking a mate, fighting rivals over food, finding a good oviposition site.

Section 9.3, I'm a bit confused by the discussion of cerebellum-like structures, because I don't think they have as dense recurrent connections as needed for the map cells in Endotaxis. Are you suggesting they are analogous to the output part of Endotaxis only, not the whole thing?

Please see our reply in the public review. We have removed this discussion of cerebellar circuits.

Line 541, "After sufficient exploration...", clarify that this is describing learning of just the output synapses, not the recurrent connections between map cells?

We have revised this entire section on the arthropod mushroom body.

In lines 551-556, the discussion is confusing and possibly not consistent with current literature. How can a simulation prove that synapses in the hippocampus are only strengthened among immediately adjacent place fields? I'd suggest either removing this discussion or adding further clarification. More broadly, the connection between Endotaxis and the hippocampus is very compelling. This might also be a good point to bring up BTSP (though you do already bring it up later).

As suggested, we removed this section.

Line 621 "The successor representation (at least as currently discussed) is designed to improve learning under a particular policy" That's not actually accurate. Ref 17 (reviewer's manuscript, cited here) is not policy-specific, and instead just learns the transition statistics experienced by the animal, using a biologically plausible learning rule that is very similar to the Endotaxis map cell learning rule (see our Appendix 5, comparing to Endotaxis, though that was referencing the previous version of the Endotaxis preprint where Oja's rule was used).

We have edited this section in the discussion and removed the reference to policyspecific successor representations.

Line 636 "Endotaxis is always on" ... this was not clear earlier in the paper (e.g. line 268, and the separation of different algorithms, and "while learning do" in Algorithm 2).

The learning rules are suspended during some simulations so we can better measure the effects of different parts of endotaxis, in particular learning vs navigating. There is no interference between these two functions, and an agent benefits from having the learning rules on all the time. The text now clarifies this in the relevant sections.

Section 9.6, I like the idea of tracing different connected functions. But when you say "that could lead to the mode switch"... I'm a bit confused about what is meant here. A mode switch doesn't need to happen in a different brain area/network, because winnertake-all could be implemented by mutual inhibition between the different goal units.

This is an interesting suggestion for the high-level control algorithm. A Lorenzian view is that the animal’s choice of mode depends on internal states or drives, such as thirst vs hunger, that compete with each other. In that picture the goal cells represent options to be pursued, whereas the choice among the options occurs separately. But one could imagine that the arbitrage between drives happens through a competition at the level of goal cells: For example the consumption of water could lead to adaptation of the water cell, such that it loses out in the winner-take-all competition, the food cell takes over, and the mouse now navigates towards food. In this closed-loop picture, the animal doesn’t have to “know” what it wants at any given time, it just wants the right thing. This could eliminate the homunculus entirely! Of course this is all a bit speculative. We have edited the closing comments in a way that leaves open this possibility.

Line 697-704, I need more step-by-step explanation/derivation.

We now derive the properties of E step by step starting from Eqn (14). The proof that leads to Eqn 14 is now in a separate article (available as a preprint and attached to this submission).

**Reviewer #3 (Recommendations For The Authors):**
Please include discussion and comparison to previous work of graph-based trajectory planning using spreading activation from the current node and/or the goal node. Here is a (far from comprehensive) list of papers that present similar algorithms:Glasius, R., Komoda, A., & Gielen, S. C. (1996). A biologically inspired neural net for trajectory formation and obstacle avoidance. Biological Cybernetics, 74(6), 511-520.Gaussier, P., Revel, A., Banquet, J. P., & Babeau, V. (2002). From view cells and place cells to cognitive map learning: processing stages of the hippocampal system. Biological cybernetics, 86(1), 15-28.Gorchetchnikov A, Hasselmo ME. A biophysical implementation of a bidirectional graph search algorithm to solve multiple goal navigation tasks. Connection Science. 2005;17(1-2):145-166Martinet, L. E., Sheynikhovich, D., Benchenane, K., & Arleo, A. (2011). Spatial learning and action planning in a prefrontal cortical network model. PLoS computational biology, 7(5), e1002045.Ponulak, F., & Hopfield, J. J. (2013). Rapid, parallel path planning by propagating wavefronts of spiking neural activity. Frontiers in computational neuroscience, 7, 98.Khajeh-Alijani, A., Urbanczik, R., & Senn, W. (2015). Scale-free navigational planning by neuronal traveling waves. PloS one, 10(7), e0127269.Adamatzky, A. (2017). Physical maze solvers. All twelve prototypes implement 1961 Lee algorithm. In Emergent computation (pp. 489-504). Springer, Cham.

Please see our reply to the public review above, and the new discussion section on “Theories and models of spatial learning”, which cites most of these papers among others.

Please explain, if it is the case, why the goal cell learning (other than a direct link between the goal and the corresponding map location) and calculation of the overlapping 'goal signal' is necessary, or at least advantageous.

Please see our reply in the public review above.

Map cells are initially introduced (line 84) as getting input from "only one or a few point cells". The rest of the paper seems to assume only one. Does it work when this is 'a few'? Does it matter that 'a few' is an option?

We simplified the text here to “only one point cell”. A map cell with input from two distant locations creates problems. After learning the map synapses from adjacencies in the environment, the model now “believes” that those two locations are connected. This distorts the graph on which the graph distances are computed and introduces errors in the resulting goal signals. One can elaborate the present toy model with a much larger population of map cells that might convey more robustness, but that is beyond our current scope.

(line 539 on) Please explain what feature in the mushroom body (or other cerebellumlike) circuits is proposed to correspond to the learning of connections in the adjacency matrix in the model.

Please see our response to this critique in the public review above. In the mushroom body, the Kenyon cells exhibit sparse responses and are recurrently connected. These would correspond to map cells in Endotaxis. For vertebrate cerebellum-like circuits, the correspondence is less compelling, and we have removed this topic from the discussion.